# Cholesterol accessibility at the ciliary membrane controls hedgehog signaling

Maia Kinnebrew[1†], Ellen J Iverson[1†], Bhaven B Patel[1], Ganesh V Pusapati[1], Jennifer H Kong[1], Kristen A Johnson[2], Giovanni Luchetti[1‡], Kaitlyn M Eckert[3], Jeffrey G McDonald[2,3], Douglas F Covey[4,5], Christian Siebold[6], Arun Radhakrishnan[2*], Rajat Rohatgi[1,7*]

[1]Department of Biochemistry, Stanford University School of Medicine, Stanford, United States; [2]Department of Molecular Genetics, University of Texas Southwestern Medical Center, Dallas, United States; [3]Center for Human Nutrition, University of Texas Southwestern Medical Center, Dallas, United States; [4]Taylor Family Institute for Innovative Psychiatric Research, Washington University School of Medicine, St. Louis, United States; [5]Department of Developmental Biology, Washington University School of Medicine, St. Louis, United States; [6]Division of Structural Biology, Wellcome Centre for Human Genetics, University of Oxford, Oxford, United Kingdom; [7]Department of Medicine, Stanford University School of Medicine, Stanford, United States

*For correspondence:
arun.radhakrishnan@
utsouthwestern.edu (AR);
rrohatgi@stanford.edu (RR)

[†]These authors contributed
equally to this work

Present address: [‡]Department
of Physiological Chemistry,
Genentech, South San Francisco,
United States

Competing interest: See
page 22

Reviewing editor: Duojia Pan,
UT Southwestern Medical Center
and HHMI, United States

**Abstract** Previously we proposed that transmission of the hedgehog signal across the plasma membrane by Smoothened is triggered by its interaction with cholesterol (Luchetti et al., 2016). But how is cholesterol, an abundant lipid, regulated tightly enough to control a signaling system that can cause birth defects and cancer? Using toxin-based sensors that distinguish between distinct pools of cholesterol, we find that Smoothened activation and hedgehog signaling are driven by a biochemically-defined, small fraction of membrane cholesterol, termed accessible cholesterol. Increasing cholesterol accessibility by depletion of sphingomyelin, which sequesters cholesterol in complexes, amplifies hedgehog signaling. Hedgehog ligands increase cholesterol accessibility in the membrane of the primary cilium by inactivating the transporter-like protein Patched 1. Trapping this accessible cholesterol blocks hedgehog signal transmission across the membrane. Our work shows that the organization of cholesterol in the ciliary membrane can be modified by extracellular ligands to control the activity of cilia-localized signaling proteins.
DOI: https://doi.org/10.7554/eLife.50051.001

## Introduction

A long-standing mystery in hedgehog (HH) signaling is how Patched 1 (PTCH1), the receptor for HH ligands, inhibits Smoothened (SMO), a G-protein-coupled receptor (GPCR) family protein that transduces the HH signal across the membrane (*Kong et al., 2019*). We and others demonstrated that cholesterol directly binds and activates SMO and proposed that PTCH1 regulates SMO by restricting its access to cholesterol (*Byrne et al., 2016*; *Huang et al., 2016*; *Luchetti et al., 2016*). Biochemical studies show that PTCH1 can bind and efflux sterols from cells (*Bidet et al., 2011*), and structural studies highlight the homology of PTCH1 to the cholesterol transporter Niemann-Pick C1 (NPC1) (*Gong et al., 2018*; *Kong et al., 2019*; *Kowatsch et al., 2019*; *Qian et al., 2019*; *Qi et al., 2019a*; *Qi et al., 2018a*; *Qi et al., 2018b*; *Zhang et al., 2018*). However, the resolution of the PTCH1 cryo-EM structures is not high enough to distinguish cholesterol from other sterol lipids as PTCH1

substrates, and PTCH1 transport activity has not yet been demonstrated in a purified system or at endogenous expression levels in cells.

A challenge to this model is presented by the fact that cholesterol constitutes up to 50% of the lipid molecules in the plasma membrane (*Colbeau et al., 1971*; *Das et al., 2013*; *Lange et al., 1989*; *Touster et al., 1970*): how can such an abundant lipid be kept away from SMO to prevent inappropriate pathway activation? Indeed, other less abundant lipids can bind and regulate SMO activity, including oxysterols, phosphoinositides, endocannabinoids and arachidonic acid derivatives (*Arensdorf et al., 2017*; *Jiang et al., 2016*; *Khaliullina et al., 2015*; *Nachtergaele et al., 2012*). Side-chain oxysterols, synthesized through the enzymatic or non-enzymatic oxidation of cholesterol, are appealing alternatives to cholesterol because of their lower abundance, higher hydrophilicity and structural similarity to cholesterol (*Corcoran and Scott, 2006*; *Dwyer et al., 2007*). To find the endogenous lipidic activator of SMO, we took an unbiased genetic approach to identify lipid-related genes whose loss influences the strength of HH signaling.

## Results

### A focused CRISPR screen targeting lipid-related genes

Using our previously described strategy (*Pusapati et al., 2018b*) to identify positive and negative regulators of the HH pathway, we conducted focused loss-of-function CRISPR screens using a custom library targeting 1244 lipid-related genes compiled by the LIPID MAPS consortium (*Supplementary file 1* provides a list of all genes and guide RNAs in the library). This CRISPR library targeted all annotated genes encoding enzymes involved in the synthesis or metabolism of lipids as well as proteins that bind or transport lipids. We used a previously characterized NIH/3T3 cell line (NIH/3T3-CG) that expresses Cas9 and GFP driven by a HH-responsive fluorescent reporter (GLI-GFP) (*Pusapati et al., 2018b*). To ensure that HH signaling in the plasma membranes of these cells would be sensitive to perturbations of endogenous lipid metabolic pathways, we minimized sterol uptake from the media by growing the entire population of mutagenized cells in lipoprotein-depleted media for one week prior to the screen and then further treating with U18666A, a drug that traps any residual lipoprotein-derived cholesterol in lysosomes (*Figure 1A*) (*Lu et al., 2015*). In the screen for positive regulators, we treated cells with a high, saturating concentration of the ligand Sonic Hedgehog (HiSHH) and used Fluorescence Activated Cell Sorting (FACS) to collect poor responders, those with the lowest 10% of GLI-GFP fluorescence (*Figure 1A and B*; full screen results in *Supplementary file 2*). These candidate positive regulators are hereafter referred to as 'HiSHH-Bot10%'. In the screen for negative regulators, we treated cells with a low, sub-saturating concentration of SHH (LoSHH) that activated the reporter to <10% of maximal strength and selected super-responders, cells with the top 5% of GLI-GFP fluorescence (*Figure 1A and C*; full screen results in *Supplementary file 3*). These candidate negative regulators are hereafter referred to as 'LoSHH-Top5%'.

The screens correctly identified all four positive controls included in the library: *Smo* and *Adrbk1* (or *Grk2*) as positive regulators and *Ptch1* and *Sufu* as negative regulators (*Figure 1B and C*). In addition, genes previously known to influence HH signaling (*Gnas*) and protein trafficking at primary cilia (*Inpp5e*) were amongst the most significant hits (*Chávez et al., 2015*; *Garcia-Gonzalo et al., 2015*; *Regard et al., 2013*). In addition to *Inpp5e*, other genes involved in phosphoinositide metabolism (*Mtmr3* and *Plcb3*) were also significant hits. *Pla2g3*, which encodes a secreted phospholipase, was identified as a negative regulator of HH signaling, an effect that may be related to its known role as a suppressor of ciliogenesis (*Figure 1D*) (*Gijs et al., 2015*; *Kim et al., 2010*).

To identify lipid species that influence HH signaling, we analyzed the intersection of all genes expressed in NIH/3T3-CG cells based on RNAseq and annotated as part of a lipid metabolic pathway in the Kyoto Encyclopedia of Genes and Genomes (KEGG) (*Figure 1D*; gene lists used for each pathway are shown in *Supplementary file 4*; RNAseq data in *Supplementary file 5*). Statistically significant hits clustered in two major pathways: (1) genes encoding enzymes in the cholesterol biosynthesis pathway were positive regulators of HH signaling and (2) genes encoding enzymes in the sphingolipid biosynthesis pathway were negative regulators (positive regulators are shown in blue

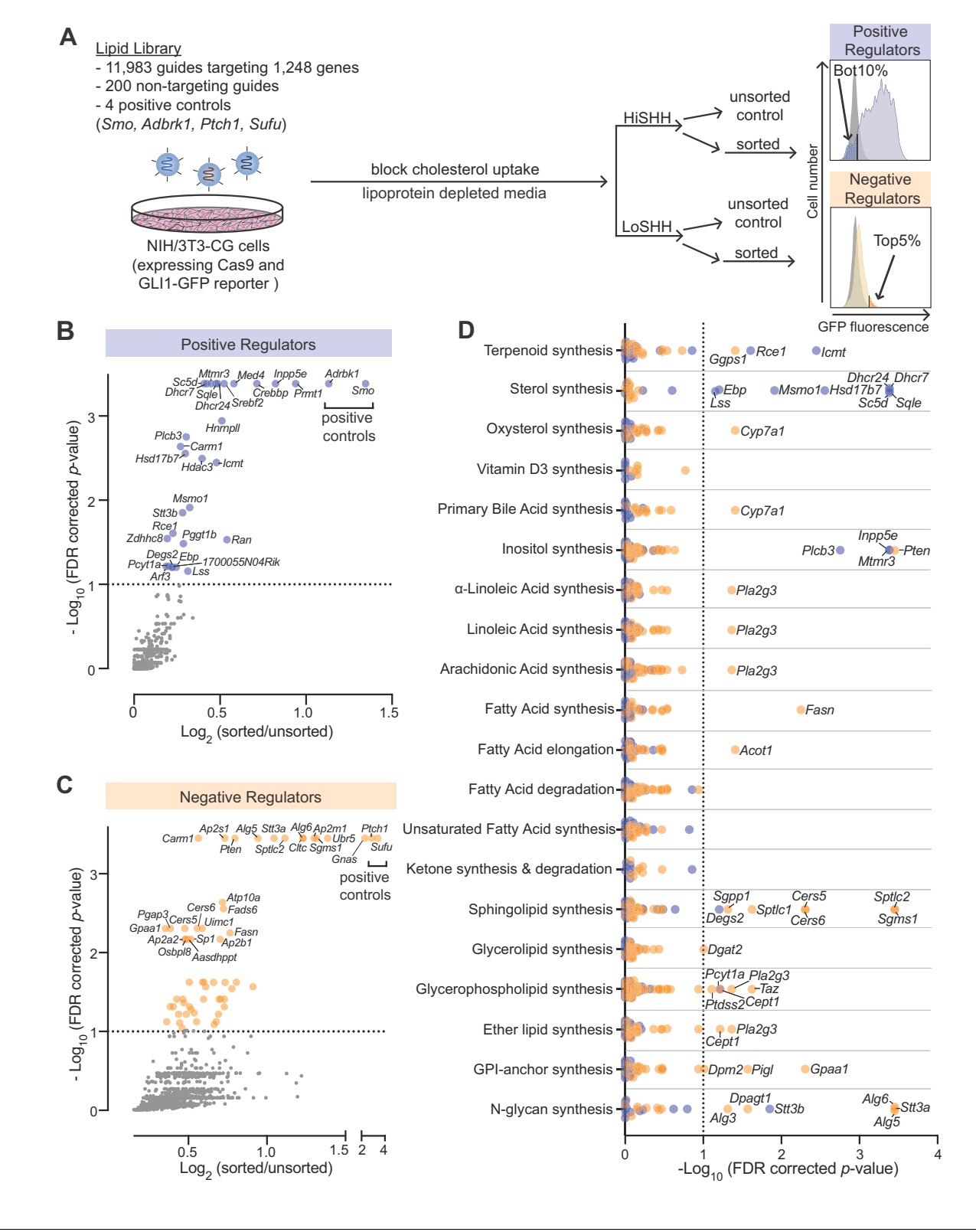

**Figure 1.** CRISPR screens identify lipid-related genes that influence hedgehog signaling. (**A**) Flowchart summarizing the screening strategy. Screens for positive and negative regulators used a high, saturating concentration of SHH (HiSHH, 25 nM) or a low concentration of SHH (LoSHH, 3.2 nM), respectively. (**B and C**) Volcano plots of the HiSHH-Bot10% (**B**) screen for positive regulators and the LoSHH-Top5% (**C**) screen for negative regulators. Enrichment is calculated as the mean of all sgRNAs for a given gene in the sorted over unsorted population, with the y-axis showing significance based

*Figure 1 continued on next page*

*Figure 1 continued*

on the false discovery rate (FDR)-corrected *p*-value. (D) Screen results analyzed by grouping genes based on the core lipid biosynthetic pathways in KEGG. In all panels, genes identified as positive and negative regulators are labeled in blue and orange respectively. See **Supplementary file 4** for the full analysis.

DOI: https://doi.org/10.7554/eLife.50051.002

and negative regulators in orange in **Figure 1D**). We focused on these two pathways for the work described in the rest of this study.

## Enzymes in the cholesterol biosynthesis pathway positively regulate hedgehog signaling

Mutations in *Dhcr7* and *Sc5d*, which encode enzymes that catalyze the terminal steps in cholesterol biosynthesis, impair HH signaling in target cells and cause the congenital malformation syndromes Smith-Lemli-Opitz and lathosterolosis, respectively (**Blassberg et al., 2016**; **Cooper et al., 2003**; **Horvat et al., 2011**; **Porter and Herman, 2011**). In addition to these genes, many of the genes encoding enzymes in the pathway that converts squalene to cholesterol were statistically significant hits with a FDR-corrected *p*-value threshold of 0.1 in the HiSHH-Bot10% screen (**Figure 2A**). Of the post-squalene cholesterol biosynthesis genes that did not meet this threshold, *Nsdhl* came close (FDR-corrected *p*-value=0.25, **Supplementary file 2**) and *Tm7sf2* is redundant (**Sharpe and Brown, 2013**), leaving *Cyp51* as the only gene that was not identified. CRISPR-mediated loss-of-function mutations in *Lss*, required for an early step in the pathway, and in *Dhcr7* and *Dhcr24*, required for the terminal steps, impaired the transcriptional induction of endogenous *Gli1* (an immediate target gene used as a measure of signaling strength) (**Figure 2B and C**; CRISPR-edited alleles shown in **Figure 2—figure supplement 1A**). Quantitative mass spectrometry measurements and intact cell staining with a cholesterol-binding probe confirmed that the abundance of cholesterol was reduced in *Dhcr7*[-/-] and *Dhcr24*[-/-] cells (**Figure 2—figure supplement 1B and C**). Conversely, the abundances of substrates for Dhcr7 and Dhcr24, 7-dehydrocholesterol and desmosterol, respectively, were elevated (**Figure 2—figure supplement 1D and E**). HH signaling in both *Lss*[-/-] and *Dhcr7*[-/-] cells, but not in *Dhcr24*[-/-] cells, could be rescued with the addition of exogenous cholesterol, pointing to cholesterol deficiency as the cause of impaired HH signaling (**Figure 2B and C**). Rescue of HH signaling defects in *Dhcr7*[-/-] cells by exogenous cholesterol has also been demonstrated previously (**Blassberg et al., 2016**). We do not yet understand the inability of cholesterol to rescue signaling in *Dhcr24*[-/-] cells.

The results of our unbiased screen highlight the importance of the endogenous post-squalene cholesterol biosynthetic pathway for HH signaling in target cells. While a simple explanation for this requirement is that cholesterol activates SMO in response to HH ligands, two additional possibilities have been discussed in the literature. First, defects in the terminal steps in cholesterol biosynthesis may lead to accumulation of precursor sterols that inhibit signaling (**Porter and Herman, 2011**). However, HH signaling defects caused by mutations in genes that control the earliest steps in the pathway (*Sqle*, *Lss*; **Figure 2A**) cannot be explained by the accumulation of inhibitory precursor sterols (at least not by intermediates in the synthesis of cholesterol from squalene). The second possibility is that cholesterol is not the product of this pathway directly relevant to HH signaling. Instead, a different molecule synthesized from cholesterol, such as an oxysterol, primary bile acid or Vitamin D derivative, is required (**Bijlsma et al., 2006**; **Corcoran and Scott, 2006**; **Dwyer et al., 2007**). However, none of the genes encoding enzymes that mediate synthesis of these metabolites (listed in **Supplementary file 4**) were identified as significant hits in the HiSHH-Bot10% screen (**Figure 1D**). A lone oxysterol synthesis enzyme (CYP7A1) was implicated in an opposite role, a negative regulator, in the LoSHH-Top5% screen (**Figure 1D**).

24(S), 25-epoxycholesterol is a unique oxysterol that is not synthesized from cholesterol but rather made through a shunt pathway that closely parallels the post-squalene cholesterol biosynthesis pathway (**Figure 2A**). Exogenously added 24(S), 25-epoxycholesterol can bind and activate SMO, so it has been proposed that this oxysterol (present at ~1% of the levels of cholesterol in cells) may be an endogenous SMO agonist regulated by PTCH1 (**Qi et al., 2019b**; **Raleigh et al., 2018**). Interestingly, DHCR24 is only used in the Kandutsch-Russell and Bloch pathways for cholesterol synthesis

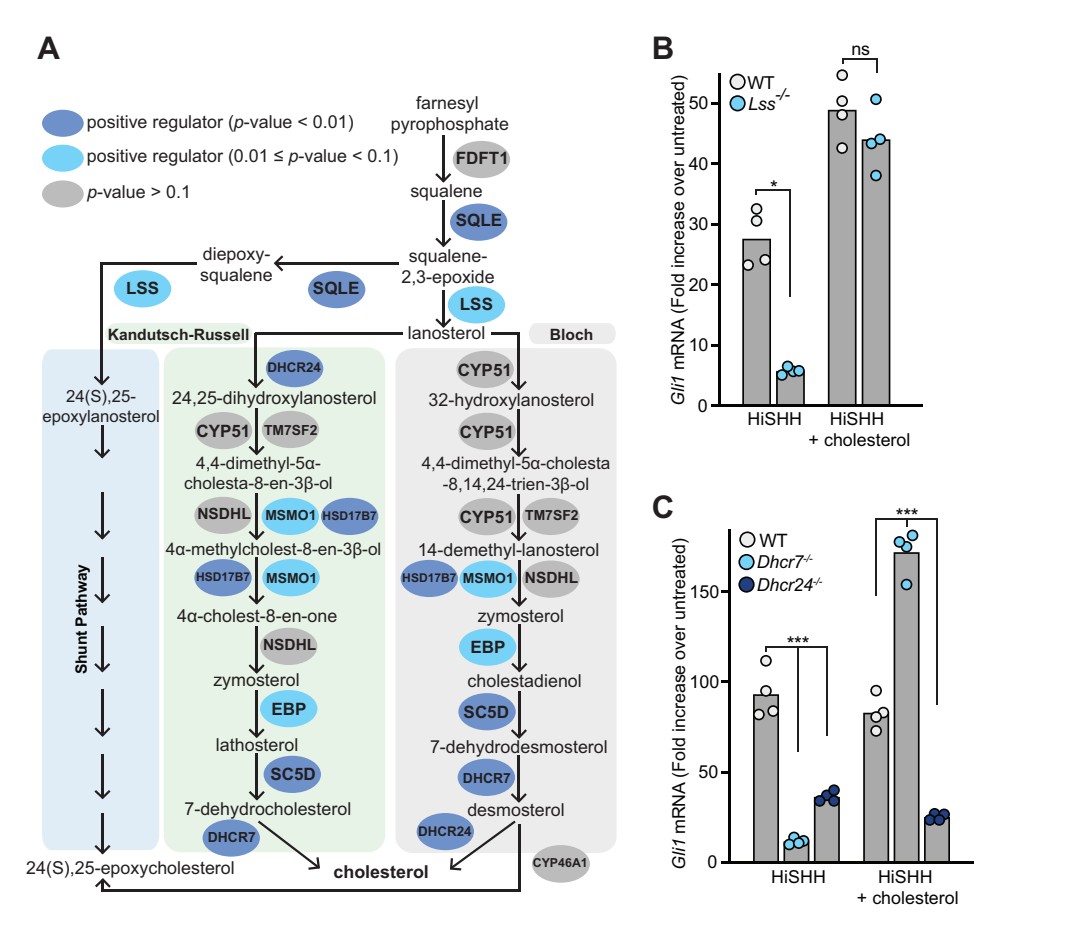

**Figure 2.** Enzymes that generate cholesterol positively regulate hedgehog signaling. (**A**) The post-squalene portion of the cholesterol biosynthetic pathway, with enzymes colored according to their FDR corrected *p*-value in our CRISPR screens (see ***Supplementary files 2*** and ***3***). Two branches of the pathway (the Kandutsch-Russell and the Bloch pathways) produce cholesterol, while a third shunt pathway produces 24(S), 25-epoxycholesterol. DHCR24 is the only enzyme that is required for cholesterol biosynthesis, but dispensable for 24(S), 25-epoxycholesterol synthesis. (**B and C**) HH signaling strength in *Lss⁻/⁻*, *Dhcr7⁻/⁻* and *Dhcr24⁻/⁻* NIH/3T3 cells was assessed by measuring *Gli1* mRNA by quantitative reverse transcription PCR (qRT-PCR) after treatment with either HiSHH (25 nM) or HiSHH combined with 0.3 mM cholesterol:MβCD complexes. Bars denote the mean value derived from the four individual measurements shown. Statistical significance was determined by the Mann-Whitney test (B, *p*-value for HiSHH treatment = 0.0286, *p*-value for HiSHH + cholesterol treatment = 0.4857) or the Kruskal-Wallis test (C, *p*-value for HiSHH treatment = 0.0002, *p*-value for HiSHH + cholesterol treatment = 0.0002).

DOI: https://doi.org/10.7554/eLife.50051.013

The following figure supplement is available for figure 2:

**Figure supplement 1.** Abundance of sterols in *Dhcr7⁻/⁻* and *Dhcr24⁻/⁻* cells.

DOI: https://doi.org/10.7554/eLife.50051.014

but not in the shunt pathway for the synthesis of 24(S), 25-epoxycholesterol (*Sharpe and Brown, 2013*). The fact that loss of DHCR24 blocks HH signaling (*Figure 2A and C*) without reducing 24(S), 25-epoxycholesterol abundance (*Figure 2—figure supplement 1F*) suggests that cholesterol rather than 24(S), 25-epoxycholesterol regulates HH signaling.

In summary, the data from our genetic screen support the view that cholesterol itself, rather than a precursor or a metabolite, is the endogenous sterol lipid that regulates SMO activation. Caveats of genetic screens include their inability to identify genes or pathways that are (1) redundant, (2) required for cell viability or growth, or (3) dependent on non-enzymatic reactions or exogenous molecules supplied by the media.

## Cellular sphingomyelin suppresses hedgehog signaling

Multiple enzymes in the sphingolipid synthesis pathway were statistically significant hits in the LoSHH-Top5% screen, indicating these enzymes are negative regulators of HH signaling strength (*Figure 3A*). Top hits from this screen include *Sptlc2*, the first committed step in sphingolipid synthesis from L-serine and palmitoyl-CoA, as well as *Sgms1*, which converts ceramide to sphingomyelin (SM). The identification of *Sgms1* suggests that SM is the relevant product of the sphingolipid pathway that attenuates HH signaling.

Because we were unable to isolate viable NIH/3T3 cell lines entirely depleted of SPTLC2 or SGMS1 protein using CRISPR editing, we used a pharmacological strategy. Myriocin is a fungal antibiotic that inhibits SPTLC2 (*Figure 3A*) and is commonly used to deplete SM in cells

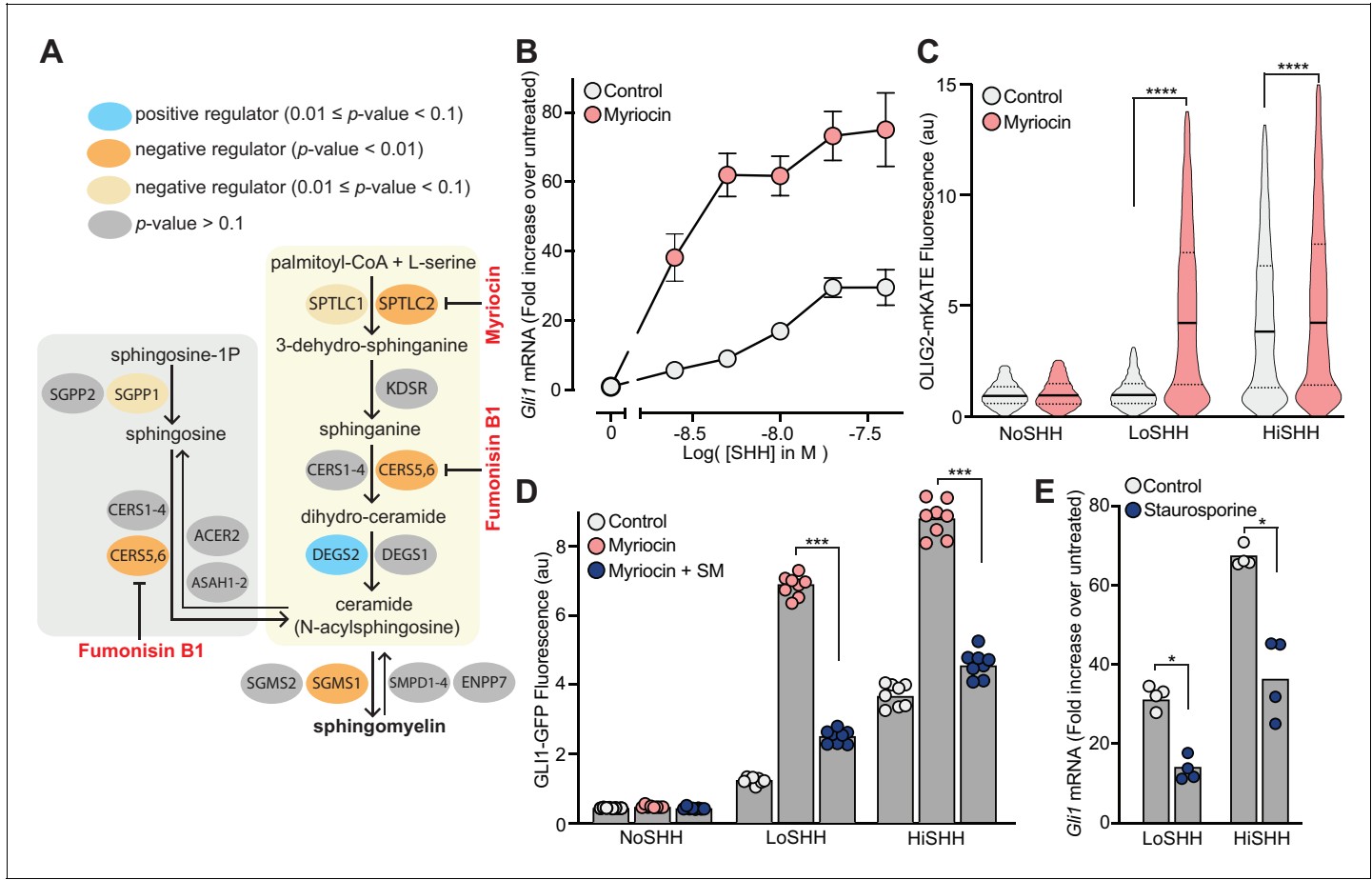

**Figure 3.** Enzymes that generate sphingomyelin negatively regulate hedgehog signaling. (A) The pathway for the synthesis of SM, with enzymes colored according to their FDR corrected *p*-value in our CRISPR screens. Steps in the pathway blocked by myriocin and fumonisin B1 are denoted in red. (B) A dose-response curve for SHH in myriocin-treated NIH/3T3 cells compared to control cells treated with vehicle (DMSO) alone. Error bars represent the standard error of the mean from four replicates. (C) Differentiation of spinal Neuronal Precursor Cells (NPCs) into OLIG2-positive motor neuron progenitors exposed to either LoSHH (5 nM) or HiSHH (25 nM) was assessed using flow cytometry to measure the fluorescence of a OLIG2-mKate reporter ($n > 5000$ cells for each treatment). (D) HH signaling strength in NIH/3T3-CG reporter cells treated with LoSHH (5 nM) or HiSHH (50 nM) after treatment with myriocin alone or myriocin followed by addition of exogenous egg SM. Each data point represents the mean GLI1-GFP fluorescence from 250 cells in two independent experiments. (E) HH signaling strength measured in NIH/3T3 cells treated with either LoSHH (5 nM) or HiSHH (25 nM) in the presence or absence of 50 nM staurosporine to increase SM. Bars denote the mean value derived from the four individual measurements shown. Statistical significance was determined by the Mann-Whitney test (C, D and E); *p*-values are: (C) *p*-value<0.0001 (both comparisons), (D) *p*-value=0.0002 (both comparisons), and (E) *p*-value=0.0286 (both comparisons).
DOI: https://doi.org/10.7554/eLife.50051.011

The following figure supplement is available for figure 3:

**Figure supplement 1.** Analysis of sphingomyelin levels and their effect on hedgehog signaling.
DOI: https://doi.org/10.7554/eLife.50051.012

(*Courtney et al., 2018*; *Tafesse et al., 2015*). SM depletion by myriocin in NIH/3T3 cells was confirmed using both thin-layer chromatography (*Figure 3—figure supplement 1A*) and flow cytometry of intact cells stained with a fluorescent protein probe (OlyA_E69A) that binds to total SM on the outer leaflet of the plasma membrane (*Figure 3—figure supplement 1B*) (*Endapally et al., 2019a*). Myriocin treatment potentiated the response to SHH in NIH/3T3 cells, as measured by the transcriptional induction of *Gli1* (*Figure 3B*). This effect was also observed in two additional cell types. In m̲ouse e̲mbryonic f̲ibroblasts (MEFs), myriocin was sufficient to activate HH signaling even in the absence of added HH ligands (*Figure 3—figure supplement 1C*). Mouse spinal n̲eural p̲rogenitor c̲ells (NPCs) differentiate into *Olig2*-expressing motor neuron progenitors in response to moderate concentrations of SHH. Myriocin potentiated the effect of SHH on NPCs, substantially reducing the concentration of SHH required to drive motor neuron differentiation (*Figure 3C*).

Several control experiments established that the potentiating effect of myriocin on HH signaling was caused by the depletion of SM, rather than an unrelated effect. First, fumonisin B1, a mycotoxin structurally distinct from myriocin that inhibits a different step in SM synthesis (*Figure 3A*), also amplified HH signaling (*Figure 3—figure supplement 1D*). Second, the potentiating effect of myriocin on HH signaling could be reversed by the exogenous administration of SM (*Figure 3D*). Finally, increasing SM levels in cells using a low-dose of staurosporine had the expected opposite effect: reduction of HH signaling strength (*Figure 3E* and *Figure 3—figure supplement 1E*) (*Maekawa et al., 2016*).

## Sphingomyelin restrains hedgehog signaling at the level of Smoothened

SM is localized in the outer leaflet of the plasma membrane (*Bretscher, 1972*; *Murate and Kobayashi, 2016*; *Verkleij et al., 1973*) where it plays key roles in its lateral organization, including the formation of ordered membrane microdomains that can influence protein trafficking, signaling and other processes (*Levental and Veatch, 2016*; *Simons and Ikonen, 2000*; *Simons and Ikonen, 1997*). Therefore, we looked broadly at the effects of myriocin on HH-relevant phenotypes, paying particular attention to primary cilia, organelles that are required for HH signaling in vertebrates (*Huangfu et al., 2003*). Myriocin did not significantly alter the abundances of the HH pathway proteins GLI3, SMO, SUFU or PTCH1 (*Figure 4—figure supplement 1A*) and also did not change either the frequency or the length of primary cilia (*Figure 4—figure supplement 1B*). Sensitivity of target cells to HH ligands can be influenced by the ciliary abundances of PTCH1, the receptor for all HH ligands that inhibits SMO, and by GPR161, a GPCR known to negatively regulate HH signaling (*Mukhopadhyay et al., 2013*; *Pusapati et al., 2018a*; *Rohatgi et al., 2007*). However, both proteins were properly localized in the ciliary membrane in myriocin-treated cells and were cleared (as expected) from cilia in response to SHH addition (*Figure 4—figure supplement 1C and D*). Thus, myriocin does not seem to significantly alter ciliary biogenesis, ciliary morphology or ciliary trafficking of receptors that negatively regulate HH signaling.

The SHH-triggered accumulation of SMO in primary cilia is required for initiation of HH signaling in the cytoplasm (*Corbit et al., 2005*). Myriocin potentiated SMO ciliary accumulation in NIH/3T3 cells, suggesting that SM depletion enhances SMO activation (*Figure 4—figure supplement 1E*). HH signaling in cells treated with myriocin was blocked by the SMO antagonist Vismodegib (*Figure 4A*). Myriocin also failed to activate HH signaling in $Smo^{-/-}$ cells (*Figure 4—figure supplement 1F*). Both observations suggest that myriocin acts on SMO, or at a step upstream of SMO. The lack of an effect of myriocin on PTCH1 trafficking (*Figure 4—figure supplement 1C*) led us to focus on SMO as the target for HH potentiation.

SMO has multiple ligand binding sites: one in the extracellular c̲ysteine-r̲ich d̲omain (CRD) that binds cholesterol and oxysterols and another in the t̲ransmembrane d̲omain (TMD) that binds to diverse SMO ligands and to sterols (*Byrne et al., 2018*; *Deshpande et al., 2019*; *Qi et al., 2019b*; *Sharpe et al., 2015*). Mutations in the CRD site abolish sterol binding and responses to SHH; mutations in the more superficial region of the TMD site prevent SMO activation by synthetic agonists like Smoothened Agonist (SAG) but do not affect responses to SHH (*Figure 4B*) (*Byrne et al., 2016*; *Huang et al., 2016*; *Luchetti et al., 2016*; *Xiao et al., 2017*). We tested the effects of these mutations on the ability of myriocin to activate HH signaling in $Smo^{-/-}$ MEFs stably expressing SMO variants. Since HH signaling is activated in these cells in response to myriocin alone (*Figure 3—figure supplement 1C*), we were able to assess the effects of these mutations without the confounding

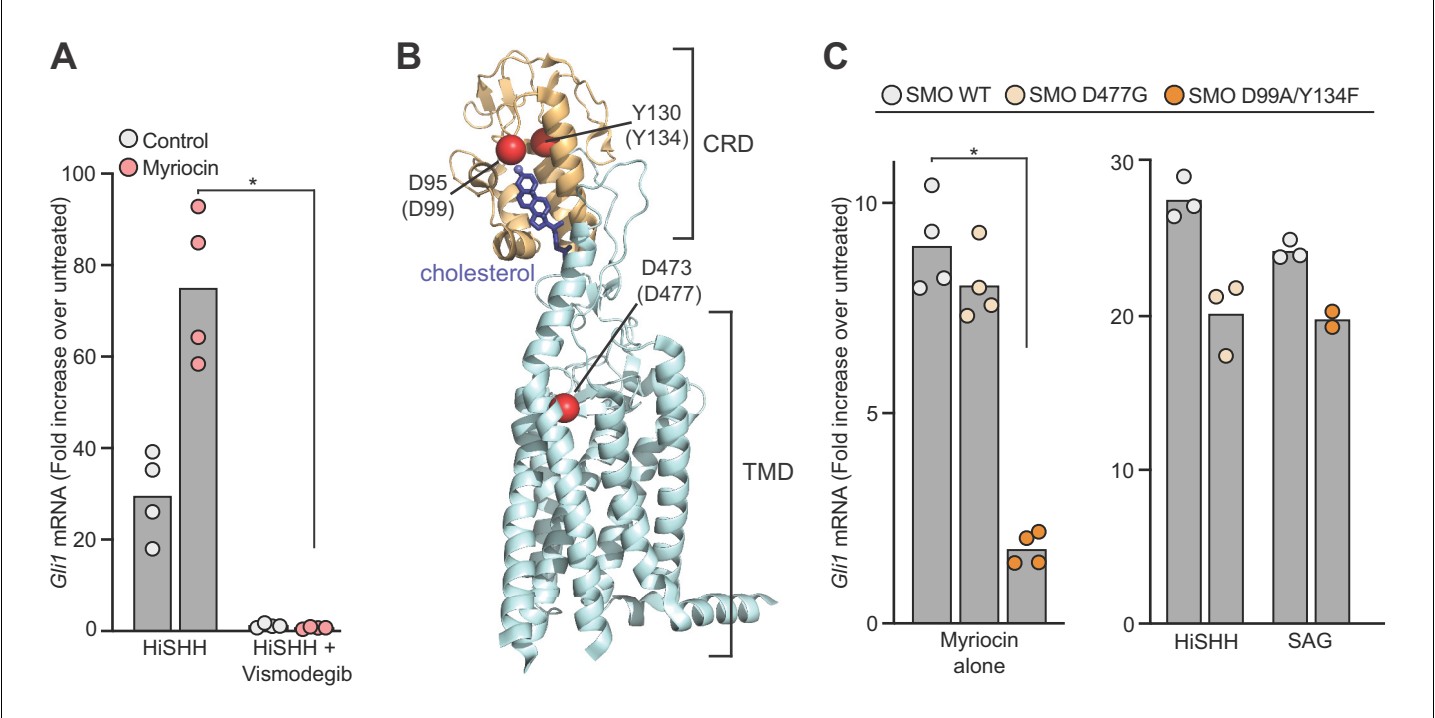

**Figure 4.** Sphingomyelin depletion potentiates hedgehog signaling at the level of Smoothened. (A) HH signaling triggered by HiSHH (25 nM) in the presence or absence of Vismodegib (2.5 μM) in NIH/3T3 cells treated with myriocin. (B) Human SMO in complex with cholesterol (PDB 5L7D) highlighting two residues in the CRD (D95 and Y130) critical for cholesterol binding and a residue (D473) in the transmembrane domain (TMD) critical for binding to the agonist SAG. Numbering for mouse SMO, used in our studies, is denoted in parenthesis. (C) HH signaling triggered by myriocin alone in *Smo*⁻/⁻ MEFs stably expressing the indicated variants of mouse SMO. A control experiment (right) shows that the SMO variants respond appropriately to either SAG (100 nM) or SHH (50 nM), demonstrating protein integrity. Note that the D477G and D99A/Y134F mutations abrogate responses to SAG and SHH, respectively (*Luchetti et al., 2016*). In (A) and (C), bars denote the mean value derived from the measurements shown (*n* = 4 for A, *n* = 2–4 in C). Statistical significance (*p*-value=0.0286 for both A and C) was determined by the Mann-Whitney test.
DOI: https://doi.org/10.7554/eLife.50051.003
The following figure supplement is available for figure 4:

**Figure supplement 1.** The effect of myriocin treatment on the abundances of hedgehog pathway components and on ciliary protein trafficking.
DOI: https://doi.org/10.7554/eLife.50051.004

effects of HH ligands or SMO agonists. Previously defined mutations in the sterol-binding CRD site (D99A/Y134F, *Figure 4B*), which abrogate cholesterol binding by disrupting a key hydrogen bond with the 3β-hydroxyl of cholesterol, reduced myriocin-driven activation (*Figure 4C*) (*Byrne et al., 2016*; *Huang et al., 2016*; *Xiao et al., 2017*). In contrast, a mutation (D477G, *Figure 4B*) in the TMD site failed to diminish myriocin-induced signaling (*Figure 4C*). In control experiments, SMO-D99A/Y134F and SMO-D477G were responsive to SAG and SHH, respectively, demonstrating protein integrity (*Figure 4C*) (*Luchetti et al., 2016*). The fact that point mutations in SMO abrogated the effect of SM depletion suggests that myriocin influences HH signaling at the level of SMO.

## Sphingomyelin restrains hedgehog signaling by sequestering cholesterol

The observation that mutations in the CRD of SMO, a well-defined binding site for cholesterol (*Byrne et al., 2016*), attenuated the effects of SM depletion (*Figure 4C*) suggested that SM regulates SMO activity by controlling the availability of cellular cholesterol that can bind to SMO. There is a precedent for SM regulation of cholesterol availability in another cellular signaling context– the control of cholesterol synthesis (*Das et al., 2014*; *Scheek et al., 1997*; *Slotte and Bierman, 1988*). These studies have led to the proposal that plasma membrane cholesterol is organized in three pools: a fixed pool essential for membrane integrity, a SM-sequestered pool with low chemical activity and a third ('accessible') pool with higher chemical activity that is available to interact with

proteins and transport to the ER (*Das et al., 2014*; *Lange et al., 2004*; *Radhakrishnan et al., 2000*). The distribution of cholesterol between the sequestered and accessible pools is determined by the ratio of cholesterol to SM: SM depletion (or cholesterol addition) leads to an increase in accessible cholesterol (*Figure 5A*) (*Das et al., 2014*). These different pools of cholesterol are characterized by differences in the chemical activity (or accessibility) of membrane cholesterol and thus cannot be measured by mass spectrometry, which extracts total cholesterol from cell membranes with solvents irrespective of the cholesterol's chemical activity. Instead, protein probes derived from lipid-binding toxins have been recently developed to distinguish between the accessible and sequestered pools in intact membranes or cells (*Figure 5A*): PFO*, derived from the bacterial toxin Perfringolysin O, binds to the accessible pool of cholesterol and OlyA, derived from the fungal toxin Osterolysin A, binds to SM-cholesterol complexes (*Das et al., 2013*; *Endapally et al., 2019a*; *Flanagan et al., 2009*; *Skočaj et al., 2014*). A useful point mutant of OlyA (OlyA_E69A) binds to both free SM and SM-cholesterol complexes, allowing measurement of total SM (*Figure 5A*) (*Endapally et al., 2019a*). To test these probes in NIH/3T3 cells, we used flow cytometry to measure the binding of fluorescently-labeled PFO*, OlyA and OlyA_E69A to intact cells either treated with myriocin to deplete SM or loaded with exogenous cholesterol (known to increase accessible cholesterol levels in cells; *Das et al., 2014*). Treatment of cells with myriocin decreased both OlyA and OlyA_E69A staining, consistent with SM depletion (*Figure 5B and C*); staining was restored by the addition of exogenous egg SM (*Figure 5D*). Myriocin treatment and cholesterol loading increased PFO* staining, showing that both treatments increased the level of accessible cholesterol in the plasma membrane (*Figure 5E*). The depletion of SM with myriocin or other pharmacological agents does not change the abundances of total cellular cholesterol or plasma membrane cholesterol (*Das et al., 2014*; *Tafesse et al., 2015*).

We sought to test the model that SM depletion by myriocin potentiates HH signaling by increasing the pool of accessible cholesterol. Reducing accessible cholesterol in myriocin-treated cells with methyl-β-cyclodextrin (MβCD), measured by PFO* staining, decreased SHH-induced activation of the GLI-GFP reporter (*Figure 5F*). This rescue is not consistent with the alternative possibility that SM negatively regulates SMO either directly or through a different mechanism.

These results, together with the requirement of the cholesterol-binding CRD for the potentiating effect of myriocin (*Figure 4C*), support the model that SM impairs HH signaling by sequestering cholesterol into complexes where it is inaccessible to SMO. This conclusion is also consistent with the observation that purified SMO is constitutively active in nano-discs containing physiological levels of cholesterol in the absence of SM (*Myers et al., 2017*). In summary, the effects of SM depletion with myriocin are reminiscent of those when cells are loaded with cholesterol: accessible cholesterol levels increase and HH signaling is potentiated (*Das et al., 2014*; *Huang et al., 2016*; *Luchetti et al., 2016*).

## Accessible cholesterol regulates hedgehog signaling

As an orthogonal approach to test the importance of accessible cholesterol without using myriocin, we used the cholesterol binding domain of the bacterial toxin anthrolysin O (ALOD4) (*Gay et al., 2015*). Unlike a reagent like MβCD, which extracts cholesterol from cells, ALOD4 selectively traps accessible cholesterol on the outer leaflet of the plasma membrane without altering cholesterol levels in the cell or plasma membrane (*Figure 6A*) (*Infante and Radhakrishnan, 2017*). Despite very different mechanisms of action, both ALOD4 and MβCD blocked HH signaling when added to cells (*Figure 6B*). In a control experiment, ALOD4 induced the expression of *Hmgcr*, the gene encoding the enzyme HMG-CoA reductase, which is known to be induced when accessible cholesterol is depleted (*Figure 6C*) (*Infante and Radhakrishnan, 2017*; *Johnson et al., 2019*). ALOD4 did not change the frequency of ciliation (*Figure 6D*).

In summary, HH signaling is enhanced by myriocin, which increases accessible cholesterol (*Figure 5E*), but inhibited by ALOD4, which decreases accessible cholesterol. Neither myriocin nor ALOD4 change total cholesterol abundance in cells (*Infante and Radhakrishnan, 2017*; *Tafesse et al., 2015*). We conclude that accessible cholesterol is the thermodynamically distinct fraction of total cholesterol that is relevant for the regulation of SMO in HH signaling.

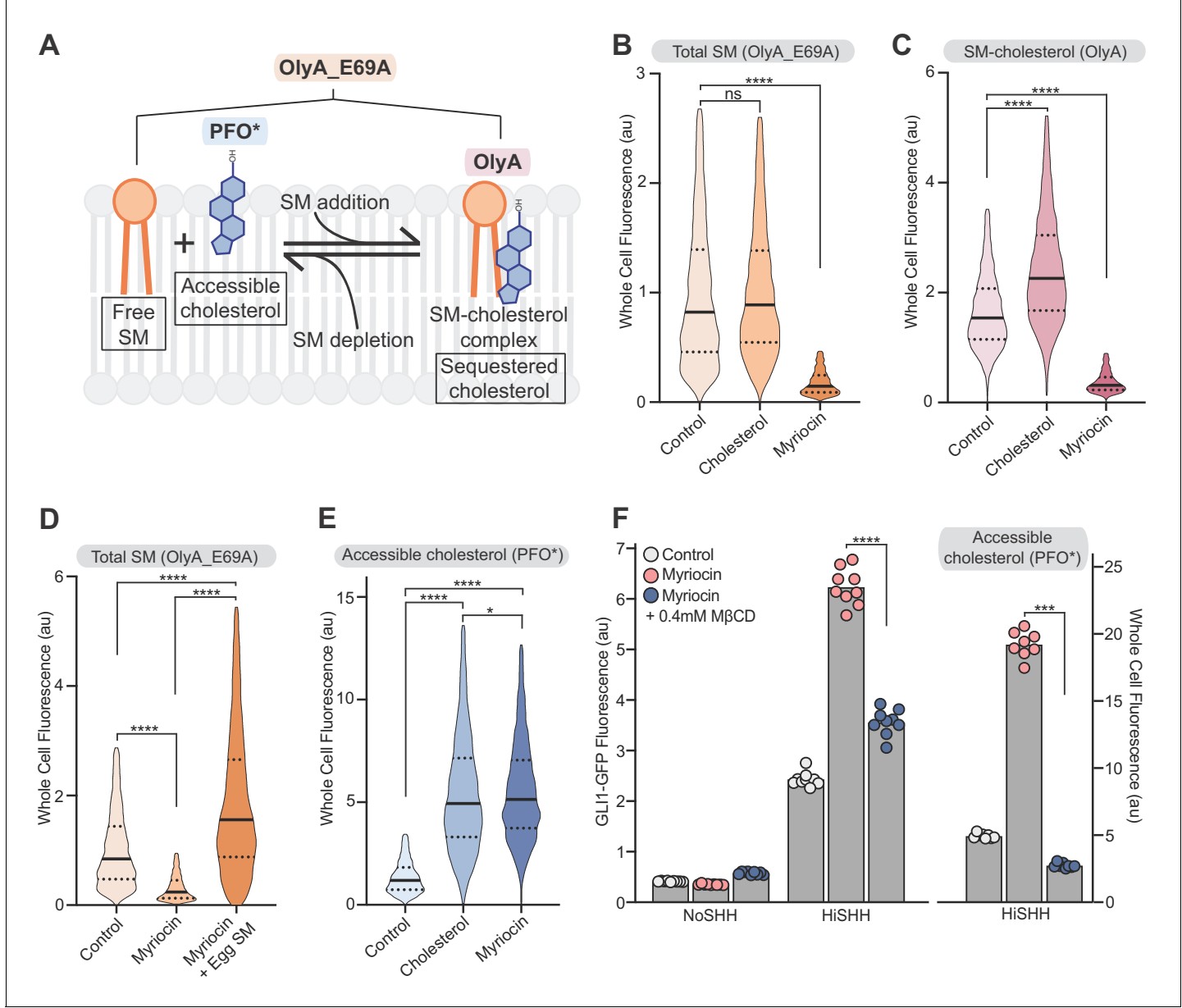

**Figure 5.** Reducing accessible cholesterol in myriocin-treated cells impairs hedgehog signaling. (A) Cholesterol and SM form SM-cholesterol complexes in which cholesterol is sequestered and prevented from interacting with proteins like SMO. The ratio of SM to cholesterol determines the level of accessible cholesterol (free from SM). Protein probes detecting the various pools of cholesterol and SM are shown: PFO* binds to accessible cholesterol, OlyA to SM-cholesterol complexes and OlyA_E69A to both free SM and SM-cholesterol complexes. (B–E) Flow cytometry of intact cells stained with fluorescently-labeled OlyA_E69A (B and D), OlyA (C) or PFO* (E) after the indicated treatments ($n$ > 4000 cells for each condition). (F) HiSHH-induced (50 nM) GLI1-GFP reporter fluorescence in NIH/3T3-CG cells treated with myriocin alone or myriocin followed by 0.4 mM MβCD to reduce accessible cholesterol levels. The graph on the right shows whole cell fluorescence of cells stained with PFO* to measure accessible cholesterol in the outer leaflet of the plasma membrane. Each data point denotes the mean fluorescence of GLI1-GFP or PFO* staining calculated from ~200 cells from two separate experiments and the bars denote the mean value. Statistical significance was determined by the Mann-Whitney test (B–F); $p$-values are: (B) $p$-value=0.9486 for control vs cholesterol treated cells and $p$-value<0.0001 for control vs myriocin treated cells, (C) $p$-value<0.0001 (both comparisons), (D) $p$-value<0.0001 (all comparisons), (E) $p$-value<0.0001 (control vs cholesterol and control vs myriocin) and $p$-value=0.0195 for cholesterol vs myriocin, (F) $p$-value<0.0001 for GLI-GFP fluorescence comparison and $p$-value=0.0002 for PFO* comparison.
DOI: https://doi.org/10.7554/eLife.50051.005

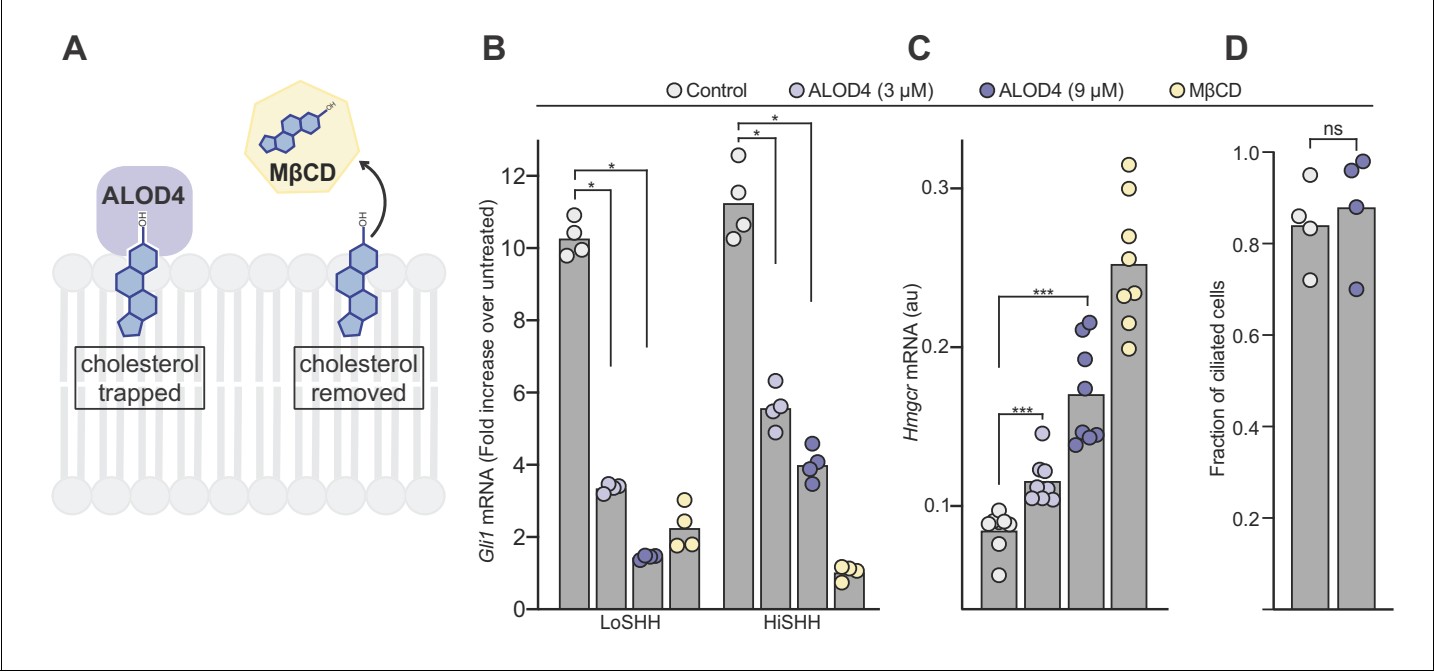

**Figure 6.** ALOD4 impairs hedgehog signaling by trapping accessible cholesterol. (**A**) ALOD4 and MβCD reduce accessible cholesterol by different mechanisms. ALOD4 binds and traps accessible cholesterol in the outer leaflet of plasma membranes of intact cells, without changing total cholesterol abundance. MβCD removes cholesterol from membranes, reducing both accessible and total cholesterol. (**B**) HH signaling triggered by five hours of LoSHH (5 nM) or HiSHH (30 nM) treatment following pre-treatment of cells with ALOD4 (3 or 9 μM) or MβCD (2 mM) for 1 hour. Bars denote the mean value derived from the four individual measurements shown. (**C**) *Hmgcr* mRNA levels measured with qRT-PCR after treatment with the same conditions as in (**B**). Bars denote the mean value derived from the eight individual measurements shown. (**D**) Ciliation frequency of cells after ALOD4 treatment (same conditions as in B), calculated as the number of cilia over the number of nuclei. Each point represents the ciliation frequency (derived from >30 cells) in a different imaging field. Statistical significance was determined by the Mann-Whitney test (**B–D**); *p*-values are: (**B**) *p*-value=0.0286 (all comparisons), (**C**) *p*-value=0.0002 for both comparisons, and (**D**) *p*-value=0.4857.
DOI: https://doi.org/10.7554/eLife.50051.006

## The ciliary membrane is a compartment with low cholesterol accessibility

The regulation of SMO by PTCH1 occurs at primary cilia, the only post-Golgi compartment in the cell where both proteins can be found localized together (*Rohatgi et al., 2009*; *Rohatgi et al., 2007*). Since the ciliary membrane is thought to have a different lipid and protein composition than the plasma membrane (*Nachury and Mick, 2019*), we compared PFO*, OlyA and OlyA_E69A staining in the ciliary membrane relative to the plasma membrane using confocal microscopy. Controls confirmed that these probes could be used to measure levels of cholesterol, SM-cholesterol complexes and total SM in the ciliary membrane using quantitative fluorescence microscopy (*Figure 7— figure supplement 1A–1C*), analogous to how we used them to measure these species at the plasma membrane by flow cytometry (*Figure 5B–5E*).

For each cilium visualized by confocal microscopy, we calculated the ratio of mean ciliary fluorescence to mean plasma membrane fluorescence in a region surrounding the cilium (hereafter the 'C/P ratio') (*Geneva et al., 2017*). We used this metric because myriocin treatment or cholesterol loading will lead to changes in probe staining at *both* the plasma membrane and the ciliary membrane. The C/P ratio reflects changes in the ciliary membrane *relative* to changes in the plasma membrane: if probe staining increases by the same factor in the plasma membrane and the ciliary membrane the C/P ratio will remain unchanged.

In untreated cells, the C/P ratio was significantly higher for OlyA_E69A staining compared to OlyA or PFO* staining (*Figure 7A*). This suggests that the ratio of SM to cholesterol, which determines the abundance of accessible cholesterol, is high in the ciliary membrane. Indeed, while OlyA_E69A staining of total SM was readily detectable in cilia, most cilia of untreated cells did not

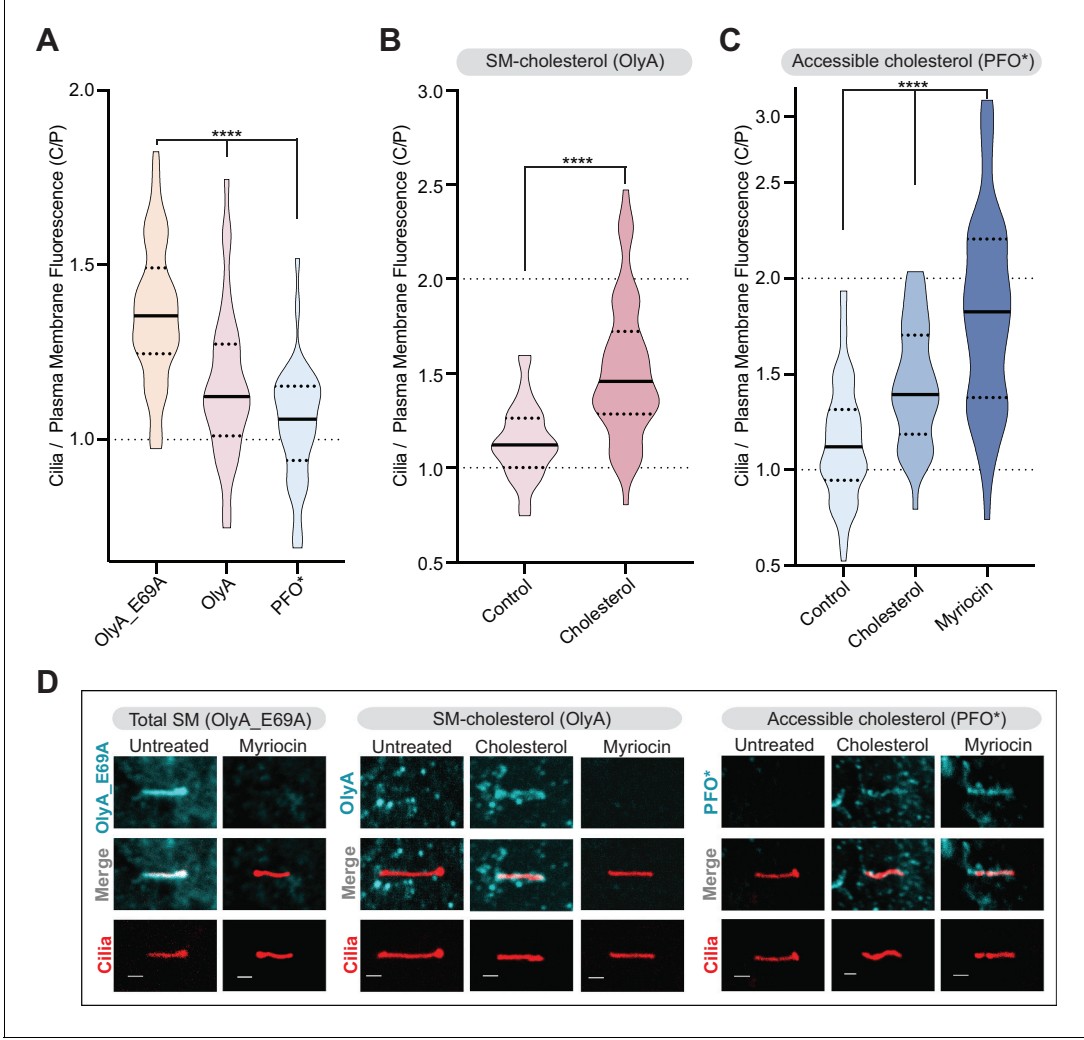

**Figure 7.** Primary cilia have high sphingomyelin and low accessible cholesterol. (A–C) Ratio of mean ciliary staining intensity to mean plasma membrane staining intensity (the C/P ratio, see text) for OlyA_E69A, OlyA, or PFO* (see *Figure 5A*) in NIH/3T3 cells left untreated (A) or treated with either myriocin (80 μM) or cholesterol-MβCD complexes (B and C) (A-C, *n* > 30 cilia per condition). (D) Representative images of individual primary cilia from cells stained with each of the lipid probes (colored blue) after the indicated treatments. Cells stably expressed ARL13B-GFP (colored red) to allow the identification of cilia. Scale bar: one micron. Statistical significance was determined by the Kruskal-Wallis test (A and C) or the Mann-Whitney test (B); all *p*-values are <0.0001.

DOI: https://doi.org/10.7554/eLife.50051.007
The following figure supplement is available for figure 7:

**Figure supplement 1.** Sphingomyelin depletion increases accessible cholesterol at cilia.
DOI: https://doi.org/10.7554/eLife.50051.008

show distinctive staining for SM-cholesterol complexes (OlyA) or accessible cholesterol (PFO*) (*Figure 7D*). Cholesterol loading increased the C/P ratio of SM-cholesterol complexes (OlyA staining, *Figure 7B and D*), showing that a significant proportion of SM molecules at cilia are free to pair with exogenously added cholesterol. Myriocin treatment, which reduced SM levels in both the plasma membrane and the ciliary membrane (*Figure 5B* and *Figure 7—figure supplement 1A*), increased the amount of accessible cholesterol in the ciliary membrane relative to the plasma membrane (PFO* staining, *Figure 7C and D*). Indeed, myriocin had a greater effect on the C/P ratio of accessible cholesterol compared to even cholesterol loading, suggesting that SM provides the major restraint on cholesterol accessibility at cilia (*Figure 7C*). Taken together, we conclude that the ratio of sphingomyelin to cholesterol is high in the ciliary membrane, leading to low cholesterol accessibility. A high ratio of SM to cholesterol provides a potential explanation for the observation that the

ciliary membrane is significantly more resistant (compared to the plasma membrane) to permeabilization by both PFO and cholesterol-binding detergents like digitonin (*Breslow et al., 2013*).

High SM levels may be critical for keeping SMO, which is cycling through the ciliary membrane even in the absence of SHH, in an inactive state by restricting its access to cholesterol (*Ocbina and Anderson, 2008*). Reducing SM levels with myriocin (or increasing cholesterol levels by cholesterol loading) increases accessible cholesterol in the ciliary membrane and, consequently, potentiates SMO activation. Conversely, reducing accessible cholesterol abundance with ALOD4 or MβCD prevents SMO activation (*Figure 6*).

## Hedgehog ligands cause an increase in cholesterol accessibility at primary cilia

PTCH1, the receptor for HH ligands, is thought to inhibit SMO by reducing its access to cholesterol using its transporter-like activity. Since PTCH1 is localized in and around the cilium, we have proposed that it could function to inhibit SMO by reducing accessible cholesterol in the ciliary membrane (*Huang et al., 2016*; *Kong et al., 2019*; *Luchetti et al., 2016*). The concept that PTCH1 can alter the membrane environment of primary cilia is also suggested by the observation that the movement and distribution of single molecules of PTCH1 and SMO in the ciliary membrane can be altered by SHH or by cholesterol depletion with MβCD (*Weiss et al., 2019*). This model predicts that SHH, which inhibits PTCH1 activity, should lead to an increase in accessible cholesterol and PFO* staining at the ciliary membrane. Given the potential artifacts associated with overexpressing a transporter protein, we sought to measure changes in accessible cholesterol at endogenous PTCH1 expression levels.

SHH did not induce much of a change in PFO* staining of the bulk plasma membrane, measured by flow cytometry (*Figure 8A*). A lack of an effect is not surprising because changes in overall cholesterol accessibility would influence many other cellular processes, including the signaling system that maintains cholesterol homeostasis (*Brown et al., 2018*; *Brown and Goldstein, 2009*; *Steck and Lange, 2010*). In contrast, SHH led to a rapid increase in accessible cholesterol in the ciliary membrane, either when cells were treated with myriocin or when cells were loaded with cholesterol (*Figure 8B and C*). SHH did not change ciliary PFO* staining in *Ptch1*−/− cells (*Figure 8—figure supplement 1A*). In addition, activation of signaling with SAG, which bypasses PTCH1 and directly activates SMO, also did not cause significant changes in accessible cholesterol at the ciliary membrane (*Figure 8—figure supplement 1B*). Both controls show that the SHH-induced change in accessible cholesterol at primary cilia is dependent on PTCH1 activity.

## Discussion

The results of our unbiased screen for lipid-related genes that influence the strength of HH signaling uncovered two pathways– cholesterol and SM synthesis– that both converge on accessible cholesterol as the critical species that regulates the interaction between PTCH1 and SMO. The potentiating effect of SM depletion on HH signaling points to cholesterol itself as the regulatory sterol, since side-chain oxysterols do not form analogous complexes with SM (and the lipid probes used in our studies do not interact with oxysterols) (*Endapally et al., 2019a*; *Slotte, 2016*). Two features explain how a seemingly abundant membrane lipid like cholesterol can play an instructive role as a second messenger in a signaling pathway. First, only a fraction of total plasma membrane cholesterol is relevant to the regulation of HH signaling. This thermodynamically distinct pool of accessible cholesterol with high chemical activity ranges from ~2% of total plasma membrane cholesterol in lipid depleted cells to ~15% of total plasma membrane cholesterol in lipid replete cells (*Das et al., 2014*). Second, the changes in accessible cholesterol that regulate SMO are confined to a subcellular compartment, the primary cilium, where PTCH1, SMO and the cytoplasmic signaling machinery downstream of SMO are localized. Phosphoinositides and oxysterols are examples of other lipids whose abundances in the ciliary membrane are different from the plasma membrane (*Chávez et al., 2015*; *Garcia-Gonzalo et al., 2015*; *Raleigh et al., 2018*), supporting the general principle that the lipid composition of primary cilia may be dynamically altered to control the activity or trafficking of cilia-localized proteins (*Nachury and Mick, 2019*). Further work will be required to determine if cholesterol accessibility is a second messenger specific to HH regulation at cilia or whether it can also influence other

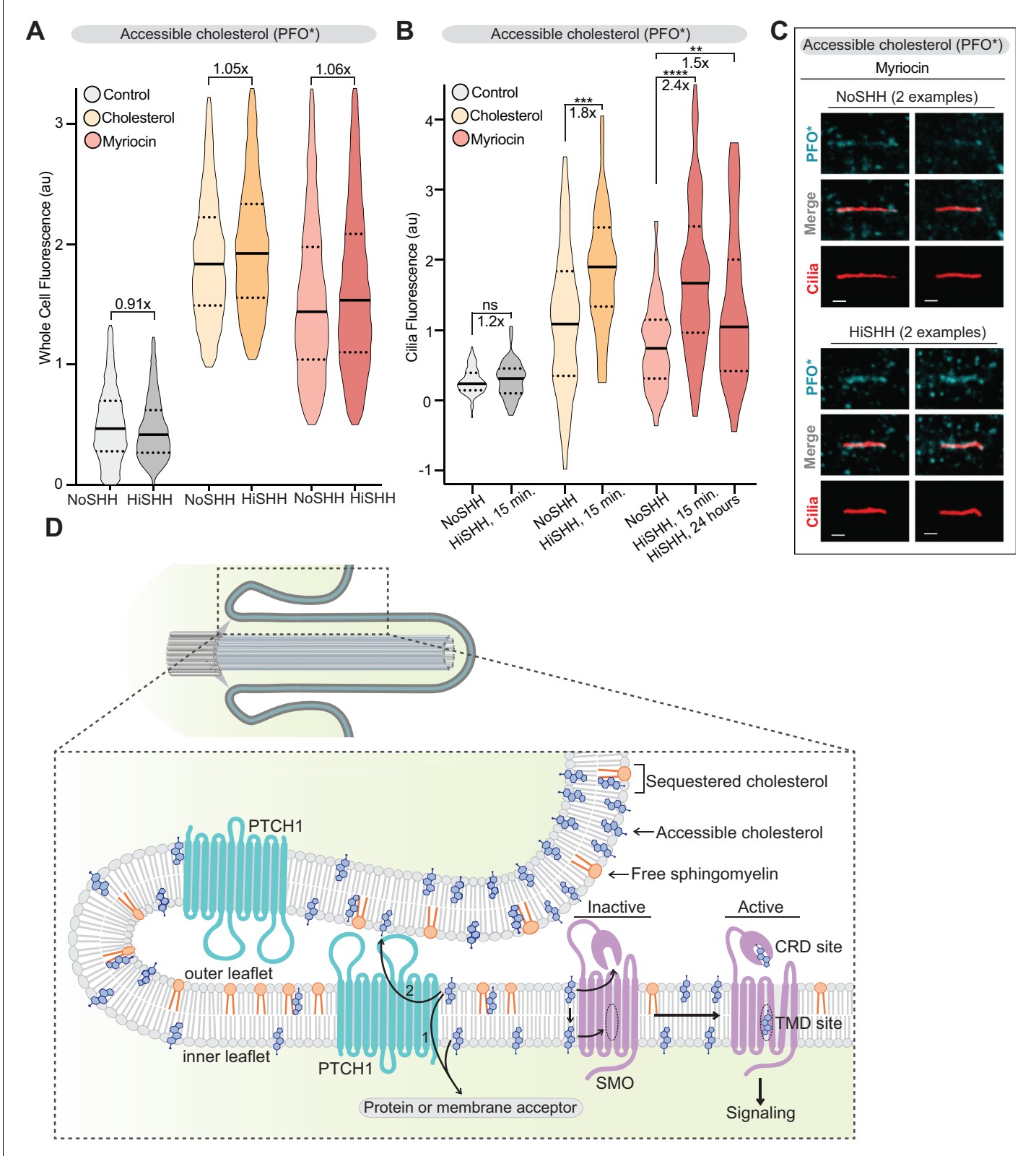

**Figure 8.** PTCH1 decreases the pool of accessible cholesterol at primary cilia. (**A**) Flow cytometry was used to measure plasma membrane PFO* staining in intact cells after HiSHH treatment (*n* > 4000 cells per violin). Fold changes of median values (SHH treated over untreated) are indicated (**A** and **B**). (**B**) PFO* staining at primary cilia after the addition of HiSHH in cells treated with myriocin (80 μM), cholesterol:MβCD complexes or left untreated (n > 65 cilia per condition). (**C**) Representative images of primary cilia from cells treated with myriocin with or without the addition of HiSHH.
*Figure 8 continued on next page*

*Figure 8 continued*

Scale bar: one micron. (D) Model depicting how PTCH1 could inhibit SMO at primary cilia by decreasing accessible cholesterol in the ciliary membrane (see Discussion for details). PTCH1 could transport cholesterol from the ciliary membrane to an intracellular acceptor (1) or to an extracellular acceptor (2). PTCH1 inactivation leads to an increase in accessible cholesterol in both leaflets of the ciliary membrane, leading to SMO activation through the CRD and TMD sterol-binding sites. Statistical significance was determined by the Mann-Whitney test (B); *p*-values are: NoSHH vs HiSHH *p*-value=0.5984, cholesterol treated NoSHH vs HiSHH *p*-value=0.0008, myriocin treated NoSHH vs HiSHH 15 min *p*-value<0.0001 and 24 hour *p*-value=0.0277.

DOI: https://doi.org/10.7554/eLife.50051.009

The following figure supplement is available for figure 8:

**Figure supplement 1.** SHH- induced changes in accessible cholesterol levels at cilia are dependent on PTCH1 activity.

DOI: https://doi.org/10.7554/eLife.50051.010

membrane proteins, including many other GPCRs, that function in the privileged compartment formed by the ciliary membrane.

The strategy of confining changes in abundance of a second messenger to a subcellular compartment or microdomain is used in other signaling pathways, such as those that use cAMP or calcium. Confinement allows small changes in absolute numbers of a regulatory molecule to be amplified into larger changes in concentration and also insulates other cellular processes from being inappropriately impacted (because second messengers are often used in multiple pathways). Another pathway that detects and responds to accessible cholesterol is the SCAP/SREBP signaling system, which provided an important precedent (and inspiration) for our work (*Brown et al., 2018*; *Brown and Goldstein, 2009*). SCAP senses accessible cholesterol in a specific subcellular compartment, the endoplasmic reticulum (ER), to regulate the transcription of genes that control cholesterol biosynthesis (*Radhakrishnan et al., 2008*; *Sokolov and Radhakrishnan, 2010*). Continual transport of cholesterol from the plasma membrane to the ER ensures that information about changes in plasma membrane cholesterol accessibility is transmitted to the ER, where SCAP is localized (*Infante and Radhakrishnan, 2017*). While cholesterol accessibility has been proposed to regulate diverse membrane proteins (*Lange and Steck, 2016*), our work provides the first clear evidence that it functions as an instructive signal in a system other than the one regulating cholesterol homeostasis.

Our work raises the important question of how PTCH1, presumably using its transporter function, reduces the levels of accessible cholesterol in the ciliary membrane (*Figure 8B*). In principle, PTCH1 could accomplish this either by increasing ciliary levels of SM or by decreasing ciliary levels of cholesterol. PTCH1 is more likely to be a sterol transporter for several reasons: it has homology to the cholesterol transporter NPC1 and the spate of recent PTCH1 structures have identified putative sterol ligands (*Gong et al., 2018*; *Qian et al., 2019*; *Qi et al., 2019a*; *Qi et al., 2018a*; *Qi et al., 2018b*; *Zhang et al., 2018*). We suggest that PTCH1 may selectively transport the pool of accessible cholesterol (rather than the fixed or SM-bound pool) to a yet unknown membrane or protein acceptor (*Figure 8D*). This would allow PTCH1 to maintain low cholesterol accessibility in a membrane compartment without depleting it entirely of cholesterol. Selective transport of accessible cholesterol has been previously proposed for other cholesterol transporter proteins (*Lange and Steck, 2016*).

The direction of sterol transfer by PTCH1 remains uncertain: it could transport cholesterol in the same direction as its relative NPC1 (from the ciliary membrane to a sterol transport protein or a membrane compartment in the cytoplasm) or in an outward direction from the ciliary membrane to an extracellular acceptor (see arrows labeled **1** and **2** in *Figure 8D*). The prominent localization of PTCH1 at and around the base of primary cilia (*Rohatgi et al., 2007*) raises the possibility that PTCH1 functions at the ciliary pocket to deplete accessible cholesterol from the ciliary membrane (*Figure 8D*). The cholesterol pumping activity of PTCH1 will be opposed by the continual leak of cholesterol back into the cilium from the large pool in the plasma membrane (since the ciliary membrane is continuous with the plasma membrane) or by the delivery of cholesterol from sterol binding proteins in the cytoplasm. This 'pump-leak' model may be energetically sustainable because PTCH1 is tasked with reducing accessible cholesterol only over the small surface area of the ciliary membrane (~400 fold smaller than that of the plasma membrane) (*Nachury, 2014*). When PTCH1 is inactivated by HH ligands, accessible cholesterol levels in primary cilia will rapidly rise due to unopposed lateral movement from the plasma membrane or delivery from cytosolic sterol transfer proteins, resulting in SMO activation.

In this model, the level of SM in the ciliary membrane determines the set-point for how much cholesterol needs to be transported by PTCH1 to reduce accessible cholesterol below the threshold required to activate SMO. Depleting SM (or loading cells with cholesterol) raises the demands on PTCH1 transport activity and hence makes cells hyper-sensitive to PTCH1-inactivating HH ligands. In some cell types (MEFs, see *Figure 3—figure supplement 1C*), PTCH1 cannot completely overcome the effect of SM depletion, leading to SMO activation even in the absence of any SHH. The emphasis in this model on the SM-cholesterol ratio accounts for the observation that either SM depletion (*Figure 3B*) or cholesterol loading (*Huang et al., 2016*; *Luchetti et al., 2016*) synergizes with HH ligands to activate signaling. Our focus on SM was driven by the results of our screen, but we note that other phospholipids can also complex cholesterol and may reduce its chemical activity at the ciliary membrane (*Keller et al., 2000*; *Lange et al., 2013*; *McConnell and Radhakrishnan, 2003*).

SMO has two potential cholesterol-binding sites: one in the CRD (tested in *Figure 4C*) and a second deep in the TMD (*Byrne et al., 2016*; *Deshpande et al., 2019*; *Huang et al., 2018*; *Luchetti et al., 2016*; *Qi et al., 2019a*). Mutations in either site prevent the activation of signaling by HH ligands, implicating both in the regulation of SMO by PTCH1. The extracellular CRD site is ~12 angstroms away from the outer leaflet of the plasma membrane (*Luchetti et al., 2016*), while the TMD site has been proposed to obtain cholesterol from the inner leaflet through an opening between two transmembrane helices (*Deshpande et al., 2019*; *Huang et al., 2018*). In the latter case, PTCH1 has been proposed to inhibit SMO by reducing cholesterol abundance in the inner leaflet of the plasma membrane (*Zhang et al., 2018*).

SM is confined to the outer leaflet of the plasma membrane and thus its depletion will directly increase accessibility of cholesterol in the outer leaflet (*Murate and Kobayashi, 2016*; *Steck and Lange, 2018*). The non-lytic sensors used throughout our work are added to intact cells and hence monitor levels of accessible cholesterol, SM, and SM-cholesterol complexes in the outer leaflet only. However, SM depletion (or cholesterol loading) will also increase the abundance of cholesterol in the inner leaflet of the plasma membrane because cholesterol rapidly redistributes between the two leaflets of the plasma membrane by flip-flop movement (*Steck and Lange, 2018*). Cholesterol flip-flop between the inner and outer leaflets also means that exogenously added ALOD4 will trap accessible cholesterol in the outer leaflet but will also consequently reduce cholesterol levels in the inner leaflet (*Infante and Radhakrishnan, 2017*). Thus, SM depletion (or ALOD4 addition) can influence cholesterol access to either the CRD or the TMD sites in SMO (*Figure 8D*). Our experiments cannot distinguish between whether inner or outer leaflet cholesterol is more relevant to SMO activation.

We end with a speculative answer to an enigma in HH signaling: why is the HH pathway dependent on primary cilia in vertebrates but not in *Drosophila*? The predominant sterol in *Drosophila* is ergosterol, with cholesterol itself representing <5% of membrane sterols (*Rietveld et al., 1999*). In addition, flies are cholesterol auxotrophs: they acquire cholesterol from their diet and have lost many of the genes for cholesterol biosynthesis (*Carvalho et al., 2010*; *Vinci et al., 2008*). Thus, flies do not need the regulatory machinery that monitors accessible cholesterol in the plasma membrane and adjusts the transcription of cholesterol biosynthetic genes. The SREBP pathway, which monitors accessible cholesterol in vertebrates, has been repurposed in *Drosophila* to respond to phosphatidylethanolamine (*Dobrosotskaya et al., 2002*). Cholesterol levels in *Drosophila* are instead sensed by a distinct nuclear receptor-based mechanism (*Bujold et al., 2010*). We propose that the lack of a need to regulate cholesterol biosynthetic pathway genes abrogates the need to confine HH signaling to primary cilia in insects.

## Materials and methods

### Key resources table

Provided in *Supplementary file 6*.

### Reagents

Suppliers for chemicals included Sigma-Aldrich (U18666A, cholesterol, Methyl-β-cyclodextrin (MβCD), fatty acid free Bovine Serum Albumin (BSA), Atto-647N maleimide); Cayman Chemicals (Myriocin, Fumonisin B1), Calbiochem (Staurosporine), Avanti Polar Lipids (Egg sphingomyelin),

Matreya LLC (Milk sphingomyelin), Enzo Life Sciences (SAG), LC Labs (Vismodegib) and Invitrogen (Alexa-647 NHS ester). Primary (SMO, PTCH1, SUFU, GLI1, GLI3 and P38) and secondary antibodies (Peroxidase AffiniPure Donkey Anti-Mouse, Rabbit and Goat IgG) used for Western Blotting were previously described (*Pusapati et al., 2018b*). Human SHH was expressed and purified as described previously (*Bishop et al., 2009*). Reagents used for cell culture are discussed in the methods for Cell Culture and Drug Treatments.

## Guide RNA library targeting lipid-related genes

In order to generate a lipid-library, human lipid-modifying genes/proteins were downloaded from the LIPID MAPS Proteome Database (https://www.lipidmaps.org/data/proteome/LMPD.php). Using Entrez IDs, these human gene names/IDs were converted to their mouse homologs using a database from the Mouse Genome Informatics (MGI). Any human genes not found in the MGI database were manually confirmed to not have a mouse homolog and excluded from the library, or, if a mouse homolog existed for the human gene, it was added to the target gene list. Missing genes were supplemented manually. This resulted in a list of 1244 mouse lipid-related genes. Finally, *Ptch1*, *Sufu*, *Smo*, and *Adrbk1* (*Grk2*) were added to the target gene list as positive controls. Using lists from the Brie Mouse CRISPR Knockout Pooled library (*Doench et al., 2016*) and the GeCKO v2 Mouse CRISPR Knockout Pooled library (*Sanjana et al., 2014*), guide RNA (sgRNA) sequences were extracted. The guide count per gene ranged from 4 to 10. Approximately 15 genes had no guides in the Brie or GeCKO libraries. For these genes, 10 guide sequences were designed using the Broad Institute's CRISPR Design tool (https://portals.broadinstitute.org/gpp/public/analysis-tools/sgrna-design). The final target gene list contained 11,783 guides targeting 1248 genes. 200 non-targeting control guides were added to this final list from the GeCKO v2 library, resulting in a total of 11,983 guides for the library. The library of guides was synthesized using Twist Bioscience's Oligo Pools and cloned into lentiGuide Puro vector (a gift from Feng Zhang; Addgene plasmid #52963) as described previously (*Joung et al., 2017*).

## CRISPR/Cas9 screen targeting lipid-related genes

Our reporter-based screening platform has been described previously in detail (*Lebensohn et al., 2016*; *Pusapati et al., 2018b*). NIH/3T3-CG cells were used because they respond to SHH in a concentration-dependent manner, carry stably integrated Cas9, and carry fluorescence-based, quantitative reporter of HH signaling (GLI-GFP) (*Pusapati et al., 2018b*). This reporter allows the isolation of cell populations with enhanced or reduced HH signaling phenotypes by FACS. CRISPR library amplification, lentiviral production, functional titer determination and transduction were carried out as previously described in detail (*Joung et al., 2017*; *Pusapati et al., 2018b*).

To prepare the library of cells for screening, $4 \times 15$ cm plates were seeded with 5 million cells each. These cells were then grown for 1 week in 'supplemented DMEM' (see Materials and methods section on Cell Culture and Drug Treatments) containing 5% Lipoprotein Depleted Serum (LDS) in place of 10% Fetal Bovine Serum. This treatment was carried out so that cells would become reliant on their own endogenous lipid-biosynthesis machinery. Finally, cells were grown to confluence in 5% LDS DMEM and then serum starved in 0.5% LDS DMEM and treated with 1 µM U18666A and either left untreated (NoSHH) or treated with LoSHH (3.2 nM) or HiSHH (25 nM) for 24 hr. Cells were then trypsinized, 4 million were pelleted and frozen for the unsorted control population, and the remaining ~25 million cells (representing 2000-fold coverage of the sgRNA library) were sorted for the lowest 10% (HiSHH-Bottom10% screen) or highest 5% of GFP fluorescence (HiSHH-Top5% screen). Finally, genomic DNA was extracted from the unsorted and sorted cells and the sgRNA library was amplified by nested PCR, subjected to Illumina sequencing, and analyzed using the MAGeCK algorithm as described previously (*Li et al., 2014*; *Pusapati et al., 2018b*).

## Kyoto encyclopedia of genes and genomes (KEGG) analysis of lipid pathways

In order to determine which lipids influence hedgehog signaling, mouse-specific genes were manually curated into lists for each lipid metabolic pathway identified on the Kyoto Encyclopedia of Genes and Genomes (KEGG) website (*Supplementary file 4*). Since oxysterol pathways are not a separate category in KEGG, manual curation of the literature was used to identify 34 oxysterol-related

enzymes (see *Supplementary file 4*), 6 of which were not found in the KEGG database (*Abdel-Khalik et al., 2018*; *Griffiths et al., 2019*; *Griffiths et al., 2017*; *Griffiths and Wang, 2018*; *Raleigh et al., 2018*; *Sever et al., 2016*). For each gene identified in KEGG or the oxysterol list, FDR-corrected *p*-values were extracted from the HiSHH-Bot10% screen (*Supplementary file 2*) as well as the LoSHH-Top5% screen (*Supplementary file 3*). Finally, the expression level of each gene was obtained from RNAseq analysis (performed in duplicate and RPKM normalized) carried out in NIH/3T3 cells (*Supplementary file 5*). If a gene was not expressed (RPKM value of 0 in both RNAseq data sets), it was not included in the pathway analysis of *Figure 1D*.

## Cell culture and drug treatments

NIH/3T3-CG Reporter Cells used in the CRISPR screen (and in *Figure 3D and 5F*), $Smo^{-/-}$ MEFs stably expressing SMO mutants (WT, D477G and D99A/Y134F), NIH/3T3 Flp-In cells stably expressing GPR161-YFP, mouse HM1 mESCs, and $Ptch1^{-/-}$ cells have been previously described and characterized (*Luchetti et al., 2016*; *Pusapati et al., 2018a*; *Pusapati et al., 2018b*; *Rohatgi et al., 2007*). NIH/3T3 Flp-In cells were purchased from Thermo Fisher Scientific, NIH/3T3 cells from ATCC, and HM1 mESCs from Open Biosystems and used at low passages for experiments. These purchased cells lines came with a certificate of authentication from the vendor and were used without further validation. Cell lines were confirmed to be negative for *Mycoplasma* infection. In the NIH/3T3 Flp-In background, $Lss^{-/-}$, $Dhcr7^{-/-}$ and $Dhcr24^{-/-}$ clonal cell lines were generated using a two-cut CRISPR strategy using methods described in our previous publications (*Pusapati et al., 2018a*; *Pusapati et al., 2018b*) and validated using a PCR-based genotyping strategy (*Figure 2—figure supplement 1A* and *Supplementary file 6*). NIH/3T3 cells and $Ptch^{-/-}$ MEFs (*Rohatgi et al., 2007*) stably expressing ARL13B C-terminally tagged with GFP were generated by lentiviral infection followed by puromycin (Calbiochem) selection.

All cells were grown in high glucose Dulbecco's Modified Eagle's Medium (DMEM) (Thermo Fisher Scientific) containing the following supplements (hereafter referred to as 'supplemented DMEM'): 10% Fetal Bovine Serum (FBS) (Sigma), 1 mM sodium pyruvate (Gibco), 2 mM L-glutamine (Gemini Biosciences), 1x MEM nonessential amino acids solution (Gibco), penicillin (40 U/ml) and streptomycin (40 micrograms/ml) (Gemini Biosciences). To induce ciliation, cells were grown to confluence and then the cell media was exchanged to low serum (0.5% FBS) supplemented DMEM. In order to test the requirement for cholesterol in hedgehog signaling, $Dhcr7^{-/-}$ and $Dhcr24^{-/-}$ NIH/3T3 cells were cultured for 1 week prior to experiments in supplemented DMEM containing 5% Lipoprotein-Depleted Serum (LDS) in place of 10% FBS (Kalen Biomedical, LLC). To induce ciliation and test for SHH responsiveness, cells were serum starved in 0.5% LDS supplemented DMEM and simultaneously treated with 1 μM U18666A and various concentrations of SHH in the presence or absence of exogenously added cholesterol:MβCD complexes for 24 hr. Due to their inability to survive with prolonged exposure to LDS-containing DMEM, $Lss^{-/-}$ cells were cultured for 2 days in 5% LDS media prior to overnight treatment with 1 μM U18666A, SHH, and cholesterol:MβCD complexes for hedgehog signaling assays. Two or more independent clonal $Dhcr7^{-/-}$, $Dhcr24^{-/-}$ and $Lss^{-/-}$ cell lines were tested in all assays and gave similar results (*Figure 2B–2C*). Note that the SHH-responsiveness of $Dhcr7^{-/-}$, $Dhcr24^{-/-}$ and $Lss^{-/-}$ cells was similar to wild-type cells when cultured in media containing lipid-replete serum.

In order to deplete cells of SM, myriocin (40 μM, unless otherwise indicated) and fumonisin B1 (40 μM) were added to cells cultured in 10% FBS DMEM for three days prior to HH signaling assays, flow cytometry, or microscopy. Low-dose staurosporine (50 nM), egg SM: fatty acid free BSA complexes (30 μM), MβCD (0.3 mM), cholesterol:MβCD complexes (0.3 mM), and SHH ligands were all added for 24 hr before analysis unless otherwise indicated. Vehicle controls (such as DMSO for Myriocin and fatty acid free BSA for SM add-back experiments) were used when appropriate. Cholesterol:MβCD complexes were generated as described previously (*Luchetti et al., 2016*). Egg sphingomyelin: fatty acid free BSA complexes were made by dissolving in Optimem (Gibco) at a molar ratio of 1000:1 (sphingomyelin:BSA) followed by water-bath sonication.

## Neural progenitor differentiation assays

To assess the effect of myriocin on hedgehog (HH) signaling in a more physiological, differentiation-based assay, we used the HM1 mESC line described previously (*Pusapati et al., 2018a*). This cell

line harbors the GLI1-Venus and OLIG2-mKate dual reporter system to evaluate the strength of HH signaling output both through *Gli1* target gene induction and the Olig2 differentiation marker for motor neuron progenitors. After growth and maintenance of mESC on feeder cells, the cells were plated on 6-well gelatin-coated CellBIND plates (Corning) at a density of 100,000 cells/well for flow cytometry analysis. Differentiation was carried out in N2B27 media (Dulbecco's Modified Eagle's Medium F12 (Gibco) and Neurobasal Medium (Gibco) (1:1 ratio) supplemented with N-2 Supplement (Gibco), B-27 Supplement (Gibco), 1% penicillin/streptomycin (Gemini Bio-Products), 2 mM L-gluta-mine (Gemini Biosciences), 40 mg/mL Bovine Serum Albumin (Sigma), and 55 mM 2-mercaptoetha-nol (Gibco)). Cells were first plated (Day 0) in N2B27 medium supplemented with 10 ng/mL bFGF (R and D). One day later (Day 1), either 40 µM myriocin or vehicle control (DMSO) was added to the culture media. On Day 2, media was replaced with bFGF-supplemented media (with or without myr-iocin) containing 5 mM CHIR99021 (Axon). On Day 3, cells were cultured in N2B27 medium contain-ing 100 nM RA (Sigma-Aldrich) (with or without myriocin) and either left untreated (NoSHH) or treated with LoSHH (5 nM) or HiSHH (25 nM). A fresh medium change with the same ingredients was done on Day 4 and Day 5. Finally, on Day 6, cells were washed with PBS and trypsinized for flow cytometry analysis. OLIG2-mKate fluorescence was measured on a FACScan Analyzer at the Stanford FACS core facility. A 561 nm laser was used for excitation and emission was collected with a filter centered at 615 nm with a 25 nm bandpass.

## Hedgehog signaling assays

*Gli1* mRNA transcript levels were measured using the Power SYBR Green Cells-to-CT kit (Thermo Fisher Scientific). *Gli1* levels relative to *Gapdh* were calculated using the Delta-Ct method (CT(*Gli1*) - CT(*Gapdh*)). The RT-PCR was carried out using custom primers for *Gli1* (forward primer: 5′-ccaagc-caactttatgtcaggg-3′ and reverse primer: 5′-agcccgcttctttgttaatttga-3′), and *Gapdh* (forward primer: 5′-agtggcaaagtggagatt-3′ and reverse primer: 5′-gtggagtcatactggaaca-3′). For analysis of GLI1-GFP in NIH/3T3-CG cells by flow cytometry, cells were harvested by trypsinization and incubated with media containing 0.5% FBS at 4°C. Cells were either analyzed by flow cytometry immediately or col-lected by centrifugation and processed for staining with the various lipid probes. All lipid probes were labeled with far-red fluorescent dyes (Alexa or Atto 647, see below), so GLI-GFP reporter fluo-rescence and probe staining could be measured simultaneously by two-channel flow cytometry. Dur-ing flow cytometry, forward and side scatter plots were used to select a live cell population largely composed of single cells and this population was then analyzed without any further gating selection. GFP fluorescence (for the GLI-GFP reporter) was measured using a Sony Cell Sorter Model SH800S (*Figure 3D*) with a 488 nm laser for excitation; emitted fluorescence was measured by first excluding wavelengths lower than 487.5 nm and higher than 561 nm and then collecting with a filter centered at 525 nm and a 50 nm bandpass. In *Figure 5F*, the GFP fluorescence was measured on a BD Acuri C6 Flow Cytometer using an 473 nm laser for excitation and a 520/30 nm bandpass filter to collect emitted light.

## Purification and labeling of lipid probes

Mutant His$_6$-tagged Perfringolysin O (PFO*) was purified as previously described (*Das et al., 2014*) and covalently labeled with Alexa Fluor 647 following the manufacturer's instructions. Expression, purification and labeling of His$_6$-tagged OlyA and His$_6$-tagged OlyA_E69A was carried out as described previously (*Endapally et al., 2019a*). Briefly, both proteins were expressed in *Escherichia coli* Rosetta(DE3)pLysS cells, purified by metal-affinity and gel-filtration chromatography and their lone cysteines labelled with Atto-647 maleimide following the manufacturer's instructions (Sigma Aldrich, product #05316). Expression and purification of His$_6$- and FLAG-tagged domain 4 of anthro-lysin O (ALOD4) was carried out as described previously (*Endapally et al., 2019b*).

## Measurement of whole-cell lipid probe staining by flow cytometry

To stain cells with these labeled probes, they were harvested for flow cytometry by trypsinization fol-lowed by quenching in ice-cold low (0.5% FBS) serum supplemented DMEM. All subsequent steps were completed on ice. Cells were spun at 1000 g and then resuspended in Probe Blocking Buffer (PBB, 1x PBS with 10 mg/mL BSA). After 10 min in PBB, cells were spun and then resuspended in PBB containing desired probes at the following final concentrations: 5 µg/mL PFO*, 2 µM OlyA and

2 µM OlyA_E69A. Cells were stained for 1 hr and then washed three times in PBB before flow cytometry.

Flow cytometry to measure fluorescence from probes bound to intact cells was carried out on a Sony Cell Sorter Model SH800S (*Figure 2—figure supplement 1B*, *Figure 3—figure supplement 1B and E*, *Figure 5B–5E*, *Figure 8A* and *Figure 5F*). Live, singlet cell populations were selected based on forward and side scatter only and analyzed without any further gating. A 638 nm laser was used for excitation, and emission was measured by first excluding wavelengths lower than 639 nm and higher than 685 nm and then collecting using a 665/30 nm bandpass filter.

## Measurement of ciliary probe staining by microscopy

Probe staining at cilia was carried out using NIH/3T3 or *Ptch*$^{-/-}$ cells stably expressing ARL13B-GFP as a cilia marker to avoid the use of detergents for permeabilization. For PFO* probe staining, the coverslips were transferred to an ice-cold metal rack and intact cells were stained with PFO* (at a final concentration of 5 µg/mL) diluted into ice cold low (0.5% FBS) serum supplemented DMEM for 30 min. For OlyA and OlyA_E69A probe staining, live cells were stained at room temperature in probe (at a final concentration of 2 µM for both OlyA and OlyA_E69A) diluted in room temperature low (0.5% FBS) serum supplemented DMEM for 10 min. After staining, cells were washed with 1x PBS and then immediately fixed in 4% PFA in 1xPBS for 10 min. Coverslips were then washed three times with 1x PBS and mounted on glass slides in ProLong Diamond Antifade Mountant (Thermo Fisher Scientific) where they cured overnight at room temperature before imaging.

## Thin layer chromatography

Cells grown in the presence or absence of myriocin (40 µM for 3 days) were harvested by washing with 4°C 1x PBS, scraping and spinning at 1000 g. Lipids were extracted using chloroform/methanol/water (2:2:1). Lipid extracts were loaded onto a TLC plate (Millipore) along with egg and milk sphingomyelin as lipid standards. The plate was run in a chloroform/acetone/methanol/acetic acid/water (6:8:2:2:1) solvent system, visualized with a 0.03% Coomassie blue G, 100 mM NaCl, 30% methanol solution, and destained with a 100 mM NaCl and 30% methanol solution (*Courtney et al., 2018*).

## Analysis of lipid probe staining at cilia by immunofluorescence

Images were obtained using a Leica TCS SP5 confocal imaging system containing a 63x oil immersion objective. Quantification of probe staining at cilia was performed using code written in MATLAB R2014b using the following steps. Leica Image Files (LIF) were converted into matrices using the bfmatlab toolbox. Following a max-z-projection, cilia were identified by applying a two-dimensional median filter followed by a high-pass user-defined threshold for signal versus noise to generate a cilia mask in the cilia channel. Any group of contiguous pixels that had signal was labeled as a 'potential cilium.' Each potential cilium was then subjected to a series of tests (measuring area, eccentricity, solidity, intensity, and length). If each test was passed, the 'true cilium' pixels were then mapped to a matrix containing the pixels in the lipid probe channel. In the lipid probe channel, each focal plane was measured independently to avoid noise caused by probe staining outside of the focal plane of a given cilium. The average intensity of pixels falling within a given cilia mask was measured for each focal plane and these values were recorded in a matrix. In order to measure the local plasma membrane fluorescence around each cilium, the cilia mask was dilated to a user-defined size, and the initial cilia mask was subtracted from the dilated cilia mask in order to create the 'plasma membrane mask.' Plasma membrane probe staining was then measured by averaging the pixel intensities within the plasma membrane mask in each focal plane. To generate the final intensity value for each cilium, the focal plane containing the highest average ciliary probe staining was normalized (either by subtraction or division when specified) to the focal plane containing the highest average plasma membrane probe staining. This MATLAB code is available on GitHub (https://github.com/mkinnebr/lipids-HH; *Kinnebrew, 2019*; copy archived at https://github.com/elifesciences-publications/lipids-HH).

## Measurement of sterol abundances by mass spectrometry

For sterol measurements, cells were grown in 6 cm plates under conditions identical to those used for the screens and for the HH signaling assays shown in *Figure 2*. Cells were extensively washed

with cold 1X PBS and harvested by scraping and centrifugation. One-tenth of each cell population was saved for genomic DNA measurement and the remainder was snap frozen in liquid nitrogen and processed for cholesterol, desmosterol, 7-dehydrocholesterol and 24(S), 25-epoxycholesterol measurements by mass spectrometry according to protocols described previously (*McDonald et al., 2012*). Genomic DNA was harvested from 1/10th of the cell population using the PureLink Genomic DNA Mini Kit (Thermo Fisher Scientific) and quantitated using the Qubit dsDNA High Sensitivity Kit (Thermo Fisher Scientific). For each sample, the sterol measurement was divided by the DNA measurement to correct for differences in cell number.

## Statistical analysis

Statistical analysis was conducted in consultation with Alex McMillan, PhD (Department of Biomedical Data Science, Stanford University School of Medicine). Data analysis and visualization were performed in GraphPad Prism 8. Model figures (*Figures 1A*, *5A*, *6A* and *8D*) and biosynthesis flow diagrams (*Figures 2A* and *3A*) were made in Adobe Illustrator CS6. Immunofluorescence images were made in Fiji-2 and the Smoothened structure (*Figure 4B*, PDB 5L7D) was generated in MacPy-MOL. RNAseq data were analyzed in Partek Flow, and analysis of the screen data was performed using a published pipeline (code available at https://github.com/RohatgiLab/BAIMS-Pipeline). Violin plots were generated with default settings in GraphPad Prism 8; outliers were excluded using the Identify Outlier function of GraphPad Prism 8 (ROUT method with a Q-score = 10%). Mean and interquartile range for the data in each violin is marked by solid and dotted lines, respectively.

All statistical analyses used non-parametric methods, which do not assume an underlying normal distribution in the data. The statistical significance of differences between two groups was determined by the Mann-Whitney test and between three or more groups by the Kruskal-Wallis test. Information about error bars, statistical tests, $p$ values and $n$ values are reported in each figure legend and were calculated using GraphPad Prism 8. All experiments included at least three independent trials with consistent results, with the exception of the CRISPR genetic screens: The HiSHH screen was performed twice and the LoSHH screen was performed once.

Throughout the paper, the numerical $p$-values for the comparisons from GraphPad Prism eight are given in the figure legends and denoted on the graphs according to the following key: *$p$-value$\leq$0.05, **$p$-value$\leq$0.01, ***$p$-value$\leq$0.001, ****$p$-value$\leq$0.0001, non-significant (ns) $p$-value>0.05.

## Data availability

The complete list of genes and guide RNAs used in the targeted CRISPR genetic screen is given in *Supplementary file 1*. The complete list of hits from the screens are given in *Supplementary files 2* and *3*. The annotated pathways compiled manually from the Kyoto Encyclopedia of Genes and Genomes (KEGG) are given in *Supplementary file 4*. RNAseq data are given in *Supplementary file 5*.

## Acknowledgements

We thank Kyle Travaglini and Onn Brandman for help with the MATLAB code for automated quantitation of probe fluorescence at primary cilia, Xiaohui Zha and Kevin Courtney for helpful discussions and protocols for sphingomyelin assays, Greg Fairn for suggesting low-dose staurosporine to elevate SM levels, Danya Vazquez for help with protein purification, and Ted Steck and Yvonne Lange for helpful comments, including discussions about cholesterol thresholds for SMO activation and PFO* binding and the 'pump-leak' model for PTCH1 function at cilia. We also thank Suzanne Pfeffer, Pehr Harbury, and Eamon Byrne for comments on the manuscript and Alex McMillan, PhD (Department of Biomedical Data Science, Stanford University School of Medicine) for expert advice on statistical analysis. CS was supported by grants from Cancer Research UK (C20724/A14414 and C20724/A26752) and a European Research Council grant (647278), RR by grants from the National Institutes of Health (GM118082 and GM106078), AR by grants from the NIH (HL20948) and Welch Foundation (I-1793), JGM in part by the NIH (HL20948), MK and EJI by pre-doctoral fellowships from the National Science Foundation, KAJ by a post-doctoral fellowship from the Hartwell Foundation, GL by a pre-doctoral fellowship from the Ford Foundation, GP by a post-doctoral fellowship from the American Heart Association (14POST20370057), and JK by a post-doctoral fellowship from the American Heart Association (19POST34380734) and a K99/R00 award from the NIH (GM13251801).

## Additional information

### Competing interests

Arun Radhakrishnan: Reviewing editor, *eLife*. The other authors declare that no competing interests exist.

### Funding

| Funder | Grant reference number | Author |
|---|---|---|
| National Institutes of Health | GM118082 | Rajat Rohatgi |
| National Institutes of Health | GM106078 | Rajat Rohatgi |
| National Institutes of Health | HL20948 | Kristen A Johnson<br>Jeffrey G McDonald<br>Arun Radhakrishnan |
| Welch Foundation | I-1793 | Kristen A Johnson<br>Arun Radhakrishnan |
| Cancer Research UK | C20724/A14414 | Christian Siebold |
| Cancer Research UK | C20724/A26752 | Christian Siebold |
| European Research Council | 647278 | Christian Siebold |
| National Science Foundation | Pre-doctoral Fellowship | Maia Kinnebrew<br>Ellen J Iverson |
| American Heart Association | 14POST20370057 | Ganesh V Pusapati |
| American Heart Association | 19POST34380734 | Jennifer H Kong |
| National Institutes of Health | GM13251801 | Jennifer H Kong |
| Ford Foundation | Pre-doctoral Fellowship | Giovanni Luchetti |

The funders had no role in study design, data collection and interpretation, or the decision to submit the work for publication.

### Author contributions

Maia Kinnebrew, Conceptualization, Software, Formal analysis, Investigation, Visualization, Methodology, Writing—original draft, Writing—review and editing; Ellen J Iverson, Jennifer H Kong, Investigation, Methodology, Writing—review and editing; Bhaven B Patel, Software, Formal analysis, Investigation, Visualization, Methodology, Writing—review and editing; Ganesh V Pusapati, Formal analysis, Investigation, Methodology, Writing—review and editing; Kristen A Johnson, Investigation, Methodology; Giovanni Luchetti, Methodology, Writing—review and editing; Kaitlyn M Eckert, Jeffrey G McDonald, Formal analysis, Investigation; Douglas F Covey, Conceptualization, Writing— review and editing; Christian Siebold, Conceptualization, Formal analysis; Arun Radhakrishnan, Conceptualization, Formal analysis, Supervision, Funding acquisition, Investigation, Writing—review and editing; Rajat Rohatgi, Conceptualization, Formal analysis, Supervision, Funding acquisition, Writing—original draft, Project administration, Writing—review and editing

### Author ORCIDs

Maia Kinnebrew ORCID https://orcid.org/0000-0002-7344-8231
Ganesh V Pusapati ORCID https://orcid.org/0000-0002-1406-2566
Christian Siebold ORCID http://orcid.org/0000-0002-6635-3621
Arun Radhakrishnan ORCID https://orcid.org/0000-0002-7266-7336
Rajat Rohatgi ORCID https://orcid.org/0000-0001-7609-8858

### Decision letter and Author response

Decision letter https://doi.org/10.7554/eLife.50051.024
Author response https://doi.org/10.7554/eLife.50051.025

## Additional files

### Supplementary files

• Supplementary file 1. List of genes and sgRNAs in the custom lipid library used for the screens in *Figure 1*. The first tab lists all the sgRNAs used in the library and the second tab lists all the lipid-related genes targeted by the library.
DOI: https://doi.org/10.7554/eLife.50051.015

• Supplementary file 2. Complete tabulated results for the HiSHH-Bottom10% screen shown in *Figure 1B*. The first tab contains a description of the columns in the second tab, which contains the scores for each sgRNA output from the MAGeCK algorithm. In the MAGeCK algorithm, gene enrichment or depletion in the sorted population (compared to the unsorted) are denoted as 'pos' or 'neg' respectively. Genes are listed in rank order according to the MAGeCK criteria.
DOI: https://doi.org/10.7554/eLife.50051.016

• Supplementary file 3. Complete tabulated results for the LoSHH-Top10% screen shown in *Figure 1C*. Organized in a manner identical to *Supplementary file 2*, but for the LoSHH-Top5% screen.
DOI: https://doi.org/10.7554/eLife.50051.017

• Supplementary file 4. Compiled lists of lipid pathway genes from the KEGG database used for the analysis in *Figure 1D*. The first tab contains lists of genes identified in each lipid biosynthesis pathway found in the KEGG database for *Mus musculus*. Column A contains a legend used throughout the spreadsheet. For each lipid biosynthesis pathway, the name of the gene is given along with the FDR-corrected *p*-values from the HiSHH-Bottom10% and LoSHH-Top5% screens and the mRNA expression level in the NIH/3T3 (see *Supplementary file 5*). The second tab contains the same analysis focused on oxysterol-related genes manually curated from the literature.
DOI: https://doi.org/10.7554/eLife.50051.018

• Supplementary file 5. Transcriptional profiling of NIH/3T3 cells using RNAseq. A list of genes from NIH/3T3 cells and their respective mRNA abundances from two independent RNAseq experiments.
DOI: https://doi.org/10.7554/eLife.50051.019

• Supplementary file 6. Key Resources Table. This file describes reagents used in this study, including (when available or applicable) the type of reagent, the designation, the source and the catalogue numbers.
DOI: https://doi.org/10.7554/eLife.50051.020

• Transparent reporting form DOI: https://doi.org/10.7554/eLife.50051.021

### Data availability

All data generated or analyzed are included in Supplementary files 1–5 in this manuscript.

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
