## [Decision Letter]

**Acceptance summary:**

Kinnebrew et al. present a follow up study to their original *eLife* manuscript where they identified cholesterol as a positive regulator of Hedgehog signalling through the receptor Smoothened (SMO). Here they address an open and challenging question in the field: how does cholesterol, a highly abundant lipid in the plasma membrane, function as an instructive messenger to activate SMO? Taking advantage of recent advances in cell-based CRISPR screens and lipid probes, the authors present evidence suggesting that SMO activation and Hedgehog signaling is mediated by a biochemically defined fraction of free accessible cholesterol that is spatially confined in a specific cellular compartment: the primary cilium (PC). The accessible cholesterol level in the PC is normally inhibited by sphingomyelin (SM), which sequesters cholesterol in SM-cholesterol complexes. By inactivating the putative cholesterol transporter PTCH, which is speculated to 'flip' free cholesterol out of the ciliary membrane to create a cholesterol-poor membrane environment, Hedgehog ligands trigger SMO activation by increasing the level of accessible cholesterol in the PC.

**Decision letter after peer review:**

Thank you for submitting your article "Sphingomyelin suppresses Hedgehog signaling by restricting cholesterol accessibility at the ciliary membrane" for consideration by *eLife*. Your article has been reviewed by three peer reviewers, and the evaluation has been overseen by a Reviewing Editor and Marianne Bronner as the Senior Editor. The following individual involved in review of your submission has agreed to reveal their identity: Stacey K Ogden (Reviewer #1).

The reviewers have discussed the reviews with one another and the Reviewing Editor has drafted this decision to help you prepare a revised submission.

Overall, the manuscript is well-written and thoughtfully presented. The data provided makes a strong case for an involvement of sphingolipid biosynthesis in regulating SMO, reinforces a role for cholesterol, and offers a novel cell biological view of how the abundant membrane lipid cholesterol can play an instructive role in Hedgehog signaling. While the reviewers consider the study of high potential significance, they feel that some new data is necessary to support the major conclusions.

Essential revisions:

1) The role of endogenous PTCH1 in maintaining the low "free cholesterol" status in cilia, or the effect of SHH in increasing "free cholesterol" in cilia, is not directly evaluated. Rather, these were inferred from experiments involving the addition of myriocin. The authors report that SHH stimulation is not changing ciliary levels of free cholesterol in the absence of myriocin treatment, as detected by PFO*. This is explained with the suggestion that levels of cholesterol aren't high enough in the absence of myriocin to be detected by the probe. Support for changes in ciliary cholesterol being the physiological mechanism by which SMO is regulated would be strengthened if the change could be observed without manipulating SM levels with myriocin. It looks like there might be some PFO D4 derivatives available that have a higher/tuneable affinity for cholesterol (Liu et al., Nat. Chem. Bio. 2017, v 13 p 268). Perhaps one of these probes could be used to look for detectable ciliary cholesterol changes in the absence of SM manipulation, with appropriate comparison of SM and cholesterol levels in the cilia of *Ptch1^-/-^* and *Ptch1^+/+^* cells, and their modulation by SHH.

2) As a control, the authors should determine whether myriocin alone affects *Gli1* expression in *Smo*^-/-^ MEFs – to unequivocally show its effects are *Smo* dependent (the mutation experiments in Figure 4C depend on this). While this was addressed with vismodegib in the presence of SHH, the magnitude of activation is very different. Also, in the Figure 4C legend, it is mentioned that the *Smo* mutants abrogate SAG and SHH responses. These data are not shown – does this sentence refer to a previous study? If so, a reference is required.

3) The authors argue away the role of other sterol biosynthetic intermediates or metabolites of the shunt pathway as physiological ligands of SMO by assuming that the lipid content of the mutant cells can be inferred from the specific enzymes mutated. Given the extensive feedback in these pathways and non-enzymatic steps, this conclusion should be backed by lipidomic analysis in the CRISPR-engineered cells to confirm the predicted changes in the different lipid species that are assumed (for example, the 24(S), 25-epoxycholesterol level in *Dhcr7^-/-^* and *Dhcr24^-/-^* cells).

---

## [Author Response]

Essential revisions:1) The role of endogenous PTCH1 in maintaining the low "free cholesterol" status in cilia, or the effect of SHH in increasing "free cholesterol" in cilia, is not directly evaluated. Rather, these were inferred from experiments involving the addition of myriocin. The authors report that SHH stimulation is not changing ciliary levels of free cholesterol in the absence of myriocin treatment, as detected by PFO*. This is explained with the suggestion that levels of cholesterol aren't high enough in the absence of myriocin to be detected by the probe. Support for changes in ciliary cholesterol being the physiological mechanism by which SMO is regulated would be strengthened if the change could be observed without manipulating SM levels with myriocin. It looks like there might be some PFO D4 derivatives available that have a higher/tuneable affinity for cholesterol (Liu et al., Nat. Chem. Bio. 2017, v 13 p 268). Perhaps one of these probes could be used to look for detectable ciliary cholesterol changes in the absence of SM manipulation, with appropriate comparison of SM and cholesterol levels in the cilia of Ptch1^-/-^ and Ptch1^+/+^ cells, and their modulation by SHH.

We agree with the reviewers that our claim that changes in ciliary cholesterol accessibility regulate SMO activation would be strengthened if we could detect these changes without manipulating sphingomyelin levels with myriocin. We now present two new experiments, neither of which use myriocin, to test the idea that SHH addition (and PTCH1 inactivation) leads to an increase in accessible cholesterol.

In a new Figure 6 and associated discussion, we use a direct functional test in live cells to demonstrate that SHH treatment leads to an increase in accessible cholesterol. For this experiment, we used ALOD4, a cholesterol-binding protein that is related to PFO* and shows the same all-or-none discrimination between accessible and inaccessible cholesterol. An important difference is that, unlike PFO*, ALOD4 is non-lytic at 37°C and hence can be used in live cells. We have previously shown that when ALOD4 is added in the extracellular medium, it can trap accessible cholesterol on the plasma membrane and prevent its transport to the Endoplasmic Reticulum to signal cholesterol excess to the cholesterol regulatory machinery (Infante and Radhakrishnan, 2017). ALOD4 accomplishes this without changing total cholesterol levels in cells or in the plasma membrane. Based on that study, a prediction is that ALOD4 should similarly be able be able to trap the accessible cholesterol produced upon PTCH1 inactivation by SHH and prevent it from triggering SMO activation. This is indeed what we observe: ALOD4 prevents SHH-induced SMO activation as measured by Gli1 mRNA levels (Figure 6B). We feel that this result supports our model that PTCH1 functions by reducing accessible cholesterol. SHH addition increases accessible cholesterol by inactivating PTCH1.

New Figure 8B and associated discussion: cholesterol loading of cells (which will increase the cholesterol:SM ratio) allows PFO* binding to cilia even in the absence of myriocin. Under these conditions, we are again able to detect a SHH-induced increase in PFO* staining at cilia. This experiment demonstrates the SHH-induced increase in accessible cholesterol at cilia in an orthogonal way, without the addition of myriocin, and thus excludes the concern that our observations are specific to the myriocin-exposed condition.

Both new experiments further support our model that accessible cholesterol increases upon SHH treatment.

Still, it would be nice to also be able to directly detect the SHH-induced increase in accessible cholesterol using a fluorescence-based staining assay. We suspect that PFO* cannot detect what is likely to be a very small increase since the pool of accessible cholesterol is maintained at low levels and any increases are rapidly dissipated to intracellular pools. We are grateful to the reviewers for pointing out the Liu et al. Nat. Chem. Biol. publication (PMID:28024150) that reports PFO-D4 mutants that may have higher affinity for cholesterol and be able to detect the small, SHH-induced increases. In carefully reading this paper, the mutations and chemical modifications introduced by Liu et al. substantially increase the hydrophobicity of PFO-D4. A consequence is that these probes are no longer selective for accessible or active cholesterol (Supplementary Figure 1E-K in their paper) over total cholesterol.

2) As a control, the authors should determine whether myriocin alone affects Gli1 expression in Smo^-/-^ MEFs – to unequivocally show its effects are Smo dependent (the mutation experiments in Figure 4C depend on this). While this was addressed with vismodegib in the presence of SHH, the magnitude of activation is very different. Also, in the Figure 4C legend, it is mentioned that the Smo mutants abrogate SAG and SHH responses. These data are not shown – does this sentence refer to a previous study? If so, a reference is required.

We thank the reviewers for suggesting this control and now report the results of this experiment in Figure 4—figure supplement 1F of the revised manuscript. Myriocin fails to activate signaling in *Smo^-/-^*MEFs, showing that its effects depend on SMO and consistent with the observation that the SMO inhibitor Vismodegib blocks myriocin-driven signaling (Figure 4A). The data that the D477G and D99A/Y134F mutations attenuate responses to SAG and SHH, respectively, were reported in our original *eLife* study (the precursor study to this Research Advance). This reference has now been added to legend accompanying Figure 4.

3) The authors argue away the role of other sterol biosynthetic intermediates or metabolites of the shunt pathway as physiological ligands of SMO by assuming that the lipid content of the mutant cells can be inferred from the specific enzymes mutated. Given the extensive feedback in these pathways and non-enzymatic steps, this conclusion should be backed by lipidomic analysis in the CRISPR-engineered cells to confirm the predicted changes in the different lipid species that are assumed (for example, the 24(S), 25-epoxycholesterol level in Dhcr7^-/-^ and Dhcr24^-/-^ cells).

In an entirely new Figure 2—figure supplement 1and associated discussion, we measured the abundances of cholesterol, desmosterol (the substrate for DHCR24), 7-dehydrocholesterol (the substrate for DHCR7) and 24(S), 25-epoxycholesterol (the shunt pathway product mentioned in this reviewer comment) by quantitative mass spectrometry in WT, *Dhcr7**^-/-^*^^ and *Dhcr24**^-/-^*^^ cells. As predicted, cholesterol levels are reduced in *Dhcr7**^-/-^*and *Dhcr24*^*-/-*^ cells. Desmosterol is elevated only in *Dhcr24*^-/-^  cells and 7-dehydrocholesterol only in *Dhcr7*^*-/-*^cells, again consistent with the known precursor-product relationships in the cholesterol biosynthesis pathway (Figure 2). Most importantly, the shunt pathway product 24(S), 25-epoxycholesterol is reduced in *Dhcr7**^-/-^* cells, but not in *Dhcr24^-/-^*cells, in agreement with the literature that DHCR24 is dispensable for the synthesis of 24(S), 25-epoxycholesterol (but required for the synthesis of cholesterol). The implications of these findings for the relative roles of cholesterol and 24(S), 25-epoxycholesterol in HH signaling are now discussed in the Results.